# T-follicular helper cells are epigenetically poised to transdifferentiate into T-regulatory type 1 cells

**Josep Garnica[1], Patricia Sole[1], Jun Yamanouchi[2], Joel Moro[1], Debajyoti Mondal[2], Cesar Fandos[1], Pau Serra[1], Pere Santamaria[1,2]***

[1]Institut D'Investigacions Biomèdiques August Pi i Sunyer, Barcelona, Spain; [2]Department of Microbiology, Immunology and Infectious Diseases, Snyder Institute for Chronic Diseases, Cumming School of Medicine, University of Calgary, Calgary, Alberta, Canada

## eLife Assessment

This study provides **important** information on pre-existing epigenetic modification in T cell plasticity. The evidence supporting the conclusions is **compelling**, supported by comprehensive transcriptional and epigenetic analyses. The work will be of interest to immunologists and colleagues studying transcriptional regulation.

*For correspondence:
psantama@ucalgary.ca

**Abstract** Chronic antigenic stimulation can trigger the formation of interleukin 10 (IL-10)-producing T-regulatory type 1 (TR1) cells in vivo. We have recently shown that murine T-follicular helper (TFH) cells are precursors of TR1 cells and that the TFH-to-TR1 cell transdifferentiation process is characterized by the progressive loss and acquisition of opposing transcription factor gene expression programs that evolve through at least one transitional cell stage. Here, we use a broad range of bulk and single-cell transcriptional and epigenetic tools to investigate the epigenetic underpinnings of this process. At the single-cell level, the TFH-to-TR1 cell transition is accompanied by both, downregulation of TFH cell-specific gene expression due to loss of chromatin accessibility, and upregulation of TR1 cell-specific genes linked to chromatin regions that remain accessible throughout the transdifferentiation process, with minimal generation of new open chromatin regions. By interrogating the epigenetic status of accessible TR1 genes on purified TFH and conventional T-cells, we find that most of these genes, including *Il10*, are already poised for expression at the TFH cell stage. Whereas these genes are closed and hypermethylated in Tconv cells, they are accessible, hypomethylated, and enriched for H3K27ac-marked and hypomethylated active enhancers in TFH cells. These enhancers are enriched for binding sites for the TFH and TR1-associated transcription factors TOX-2, IRF4, and c-MAF. Together, these data suggest that the TR1 gene expression program is genetically imprinted at the TFH cell stage.

## Introduction

Interleukin 10 (IL-10)-producing regulatory T-cells (Tregs) play a central role in the maintenance of normal immune homeostasis. Tregs include the well-characterized FOXP3+ subset as well as a FOXP3-negative CD4+ T-cell type that secretes IL-10 and low levels or no IL-4 and co-expresses CD49b, LAG-3, inducible T-cell costimulator (ICOS), and/or CCR5 and PD-1 among others (*Roncarolo et al., 2018*).

We have shown that systemic delivery of nanoparticles (NPs) coated with mono-specific peptide-major histocompatibility complex class II (pMHCII) molecules (*Singha et al., 2017*) triggers the expansion and re-programming of cognate splenic T-follicular helper (TFH) CD4+ T-cells into expanded pools of transitional (referred to as TR1.1 or TR1-like) and terminally differentiated TR1 cells (referred to as TR1.2 or TR1) that resolve inflammation in various organ-specific autoimmune disease models in a disease-specific manner without impairing normal immunity (*Clemente-Casares et al., 2016*; *Umeshappa et al., 2020*; *Umeshappa et al., 2019*). These events result from the sustained assembly of large TCR microclusters, and rapid, robust, and prolonged TCR signaling on TFH cells (*Singha et al., 2017*; *Solé et al., 2023b*; *Solé et al., 2023a*). The antigen-specific TFH and TR1 sub-pools arising in response to pMHCII-NP therapy have nearly identical clonotypic composition but alternative functional properties and transcription factor (TF) expression profiles (*Solé et al., 2023b*). In addition, pMHCII-NPs trigger cognate TR1 cell formation in TFH cell-transfused immunodeficient hosts, and T-cell-specific deletion of two master regulators of TFH cell genesis (*Bcl6* or *Irf4*) blunt both pMHCII-NP-induced TFH expansion and TR1 formation. In contrast, deletion of *Prdm1* selectively abrogates the TFH-to-TR1 conversion. Together, these data indicated that TFH cells can differentiate into TR1 cells in vivo and that Blimp-1 is a gatekeeper of this cell re-programming event (*Solé et al., 2023b*). The work described herein was initiated to test the hypothesis that pMHCII-NP-induced conversion of TFH cells into TR1 cells involves epigenetic re-programming of the TFH precursors.

Chromatin can be modified at various levels, including alterations in chromatin accessibility to expose or shield specific genes from the gene expression machinery, through histone modifications that either promote or repress gene expression, or via changes in DNA methylation, typically hypomethylation. Although such processes have been shown to contribute to T-cell differentiation, such as during Th1/Th2 cell specification (*O'Garra and Arai, 2000*), they can also occur upon T-cell activation (*Avni et al., 2002*) or in response to environmental cues, such as cytokine stimulation (*Ostuni et al., 2013*). Likewise, it has been established that DNA methylation, histone modifications, and chromatin accessibility regulate T-cell activation and effector and memory responses during immune responses to infection. Effector T-cell-specific genes are demethylated and gain chromatin accessibility and naïve T-cell-associated genes are repressed. When these cells need to become memory, the naïve genes required for survival are demethylated (*Belk et al., 2022*). Another notable example of chromatin remodeling is in exhausted T-cells, where numerous effector genes, such as *Ifng*, become closed, and acquire open chromatin at genes that are upregulated in these cells, such as *Pdcd1* (*Belk et al., 2022*).

Several notable examples of redistribution of histone marks during T-cell differentiation have been described. For example, Th1 differentiation results in H3K4 di-methylation and H3 and H4 acetylation and in the creation of chromatin accessible regions at regulatory elements within the *Ifng* locus, as well as in the loss of H3K27me3 through the locus (in addition to DNA demethylation of the locus) (*Hatton et al., 2006*; *Schoenborn et al., 2007*). In contrast, activation of the Th2 program leads to the loss of permissive histone modifications in *Ifng*, addition of repressive H3K27me3 marks along the locus and DNA methylation (*Schoenborn et al., 2007*; *Jones and Chen, 2006*; *Chang and Aune, 2007*). Th2 differentiation from naïve precursors also involves the acquisition of permissive histone modifications in *Il4, Il5, Il13* and the locus control region (LCR), and the loss of H3K27me3 marks (*Avni et al., 2002*; *Koyanagi et al., 2005*). Likewise, the promoters and eight gene regulatory elements (GREs) of *Il17a* and *Il17f* genes in naïve CD4+ T-cells acquire H3K27ac marks upon culture in Th17-polarizing conditions (*Akimzhanov et al., 2007*). In memory T-cells, which display faster and greater levels of gene transcription than their naïve and effector counterparts, lineage-specific cytokine genes retain the positive histone modifications on their proximal and distal GREs that were acquired during differentiation of their precursor T-cells (e.g. H3K27ac and H3K9ac marks), even when expression of the genes ceases upon memory cell conversion, allowing rapid reactivation of these loci upon antigen re-encounter (*Fann et al., 2006*). Likewise, naive CD8+ T cells lose repressive H3K27me3 marks during the primary immune response against infections, allowing rapid upregulation of *Ifng* and *Gzmb* expression by their memory CD8+ T-cell counterparts (*Araki et al., 2009*). Overall, memory T-cells appear to retain the epigenetic signatures of their effector progenitors, thus allowing a quicker and more efficient response in subsequent antigen encounters.

There are also several noteworthy examples of changes in DNA methylation in T-cells. For example, the *Cd4* locus is hypermethylated in CD4–CD8–, CD4+CD8+ thymocytes and mature CD8+ T cells, but is demethylated in CD4+CD8–cells (*Teghanemt et al., 2022*). Likewise, in naïve T-cells, *Il4, Il5,*

*Il13,* and the corresponding LCR are hypermethylated and *Ifng* is hypomethylated (*Fields et al., 2004*; *Lee et al., 2002*). Whereas this methylation pattern is maintained during the Th0-Th1 differentiation process, it undergoes dramatic changes during Th2 formation, such that *Ifng* becomes hypermethylated, and *Il4, Il5, Il13,* and the LCR demethylate key GREs (*Fields et al., 2004*; *Lee et al., 2002*; *Kim et al., 2007*), in addition to acquiring the type of permissive histone modifications discussed above (*Avni et al., 2002*; *Koyanagi et al., 2005*). Likewise, TFH formation from naïve T-cell precursors involves demethylation of BCL-6 binding sites (*Liu et al., 2016*). In Foxp3+ Tregs, the expression of *Foxp3* and genes coding for some Treg-function-associated molecules, such as *Ctla4* and *Il2ra*, but not genes coding for TFs controlling other cell fates or cytokine genes that are repressed in Foxp3+ Tregs, are also associated with DNA demethylation (*Liu et al., 2010*; *Ohkura et al., 2012*). The development of memory T-cells also involves progressive demethylation of promoter-distal GREs of key genes (*Durek et al., 2016*).

Collectively, the above observations indicate that, in most cases, acquisition of new functional states by peripheral T-cells, including their differentiation into different T-cell subsets, involves extensive and diverse modifications of the chromatin around specific loci, although global (i.e. genomewide) epigenetic changes during these processes were not always explored. Here, we show, through a broad combination of transcriptomic and epigenetic studies, that the TFH-to-TR1 transdifferentiation process does not follow this pattern. Specifically, we find that conventional TFH cells are epigenetically poised to differentiate into TR1 cells, and that the TFH-to-TR1 transdifferentiation process is associated with extensive contraction of the chromatin and the upregulation of genes that are already epigenetically poised for expression, yet are silent, at the TFH cell stage, such as *Il10*. These genes are closed and hypermethylated in Tconv cells, but are accessible, hypomethylated, and enriched for H3K27ac-marked and hypomethylated active enhancers in TFH cells. These data suggest that genomic imprinting is a key enabler of the TFH-TR1 cell transdifferentiation process, as documented in non-immune cell types (*Holliday and Pugh, 1975*; *Muramoto et al., 2010*; *Raas et al., 2022*).

## Results

### Single-cell multiomic profiles of pMHCII-NP-induced TFH and TR1 cells

The compaction status of the chromatin has a direct impact on gene expression, by modulating the accessibility of TF binding sites and the physical interactions between GREs. We therefore sought to map changes in the genome-wide distribution and location of open chromatin regions (OCRs) along the TFH-TR1 pathway and determine whether pMHCII-NP-induced TR1-like cells inherit their chromatin exposure status from TFH cell precursors. We focused on the BDC2.5mi/I-A$^{g7}$ Tet+ cells arising in BDC2.5mi/I-A$^{g7}$-NP-treated nonobese diabetic (NOD) mice. BDC2.5mi/I-A$^{g7}$-specific CD4+ T-cells comprise a population of autoreactive T-cells that contribute to the progression of spontaneous autoimmune diabetes in NOD mice. The size of this type 1 diabetes-relevant T-cell specificity is small and barely detectable in untreated NOD mice, but treatment with cognate pMHCII-NPs leads to the expansion and formation of anti-diabetogenic TR1 cells that retain the antigenic specificity of their precursors (*Clemente-Casares et al., 2016*). As a result, treatment of hyperglycemic NOD mice with these compounds results in the reversal of type 1 diabetes (*Clemente-Casares et al., 2016*).

Since the pMHCII-NP-induced Tet+ cell pools contain TFH and TR1 sub-pools (~30% and 70%, respectively) (*Solé et al., 2023b*; *Solé et al., 2023a*), we addressed this question by analyzing the single-cell Multiome (scMultiome) (scATACseq+scRNAseq) profiles of BDC2.5mi/I-A$^{g7}$-NP-induced Tet+ cells. Whereas scRNAseq provides information about the transcriptional identity of the various cellular sub-pools, scATACseq (assay for transposable-accessible chromatin using sequencing) reveals locations of accessible chromatin at regulatory and non-regulatory regions (*Buenrostro et al., 2013*) at each cell differentiation stage along the TFH-TR1 pathway. Furthermore, to ascertain whether the pMHCII-NP-induced Tet+ TFH cells and conventional (i.e. vaccine-induced) TFH cells are equivalent not only at the transcriptional level but also at the level of chromatin accessibility, we also compared the scMultiome profiles of pMHCII-NP-induced Tet+ cells with those corresponding to: (1) purified keyhole limpet hemocyanin (KLH)-induced TFH (CD4+CD44+CXCR5$^{high}$PD1$^{high}$) cells from immunized NOD mice, which are transcriptionally identical to the Tet+ TFH cell pools from BDC2.5mi/I-A$^{g7}$-NP-treated NOD mice (*Solé et al., 2023b*; *Solé et al., 2023a*); and (2) TH0 (CD4+CD44−CXCR5−PD1−) (naïve) cells obtained from KLH-immunized animals as a control, herein referred to as Tconv cells. We

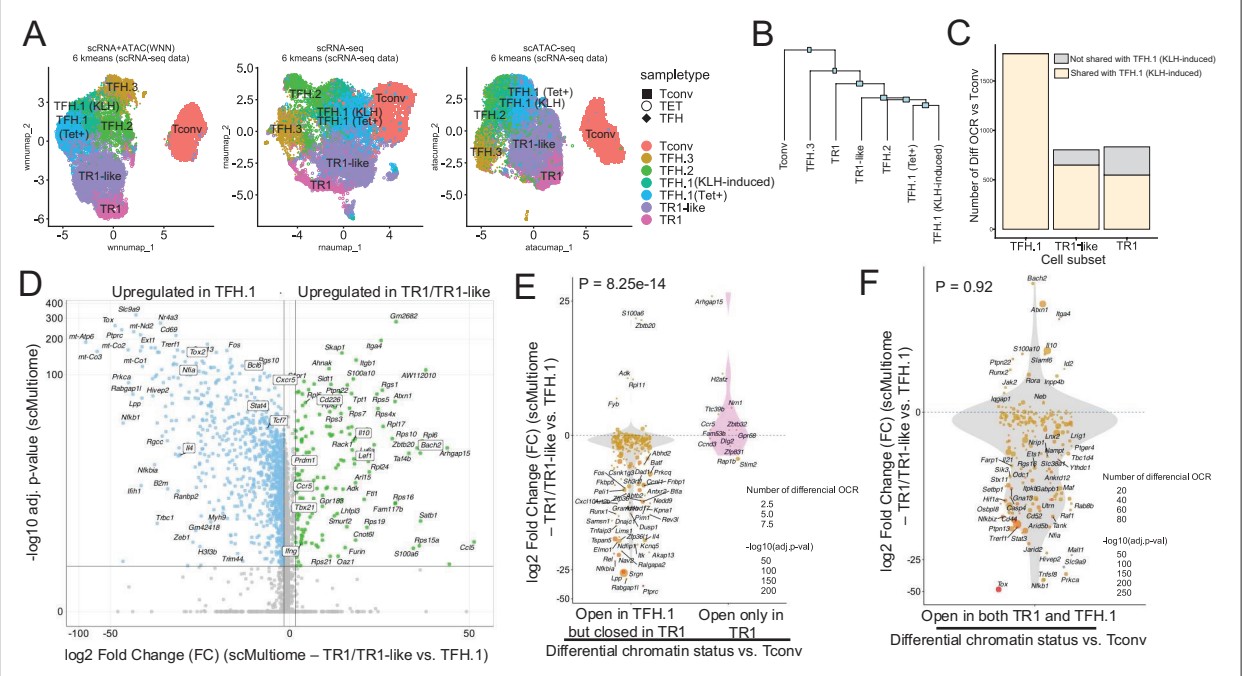

**Figure 1.** The TFH-to-TR1 conversion is associated with massive closure of previously open chromatin regions (OCRs) and gene silencing. BDC2.5mi/I-A^g7-NP-induced Tet+ (TET); KLH-DNP-induced TFH (TFH), and Tconv cells (Tconv) from female nonobese diabetic (NOD) mice (n=4, 8, and 1 mice, respectively) were sorted and processed for 10X GEX+ATAC multiome. (**A**) Left: multidimensional analysis of scRNAseq and ATACseq data using weighted nearest neighbor (WNN). Cell-type prediction is based on K-means clusterization and predicted cell subtype classification of scRNAseq data. Middle and right: scRNAseq (middle) and scATACseq (assay for transposable-accessible chromatin using sequencing) (right) UMAP (Uniform Manifold Approximation and Projection) plots with K-means clusterization and predicted cell subtype classification based on scRNAseq data. The legends' colors correspond to different T-cell types and the legends' shape correspond to sample type. (**B**) Hierarchical clustering of multiome data based on Euclidean distance of all the clusters found in all sample types. (**C**) Bar plot with the number of differentially OCRs in TFH.1 (including Tet+ TFH.1 and KLH-DNP-induced TFH.1 cells), TR1/TR1-like cells, as compared to Tconv cells (adjusted p<0.05). Color depicts the proportion of OCRs that are shared with KLH-DNP-induced TFH.1 (vs. Tconv cells) (yellow) or that appear de novo in TR1/TR1-like cells (gray). (**D**) Volcano plot of Wilcox differential analysis of scRNAseq data from the multiome dataset comparing TFH.1 and TR1/TR1-like cells. Not overlapping dots were labeled. Genes mentioned in the text are boxed. (**E–F**) Jitter plots depicting log2FC in gene expression between TR1/TR1-like and TFH.1 cells as measured by scRNAseq (adjusted p<0.05) for genes associated with OCRs found in TFH.1 cells (left; closed in TR1/TR1-like cells) or TR1/TR1-like (right; closed in TFH.1 cells) (**E**), or in both TFH.1 and TR1/TR1-like cells (**F**) as measured by scATACseq (adjusted p<0.05). Dot (gene) colors define the -log10 (adjusted p) for RNA expression of Wilcox test; dot sizes are proportional to the number of differential OCRs associated with each gene. All genes are labeled except when overlapping. Chromatin closure by TR1 cells was significantly associated with gene downregulation as determined via Chi-square test. TFH, T-follicular helper; TR1, T-regulatory type 1; KLH, keyhole limpet hemocyanin.

The online version of this article includes the following figure supplement(s) for figure 1:

**Figure supplement 1.** High transcriptional similarity between pMHCII-NP- and vaccine-induced TFH.1 cells.

note that studies of Tet+ T-cell pools before and after treatment are not possible as the frequency of Tet+ cells in the absence of treatment is below the level of detection via flow cytometry.

Cell clustering using the pMHCII-NP-induced Tet+ cell pool's two-dimensional scRNAseq and scATACseq dataset analysis using weighted nearest neighbor (WNN) revealed the presence of a well-defined TFH-like cell cluster and a larger cluster of cells containing both TR1-like and TR1 cells (*Solé et al., 2023b*). The KLH-induced TFH subset contained three sub-pools of TFH cells that we referred to as TFH.1, TFH.2, and TFH.3 (*Figure 1A*). The scMultiome profile of the BDC2.5mi/I-A^g7-NP-induced TFH cell sub-pool overlapped with the KLH-induced TFH1.1 sub-pool, corresponding to effector *Bcl6^hi Tox2^hi Il21^+ Pdcd1^+* TFH cells (*Figure 1A*, left). At the scRNAseq level, the KLH-induced TFH.2 cells were also similar to their TFH.1 counterparts but expressed lower levels of *Pdcd1*, *Il21*, *Bcl6*, and *Tox2* and higher levels of *Maf*, *Tcf7*, *Cxcr5*, and *Cd69* (*Solé et al., 2023b*). In contrast, the KLH-induced TFH.3 cells display transcriptomic features of follicular T-regulatory cells (*Foxp3^+ Bcl6^+ Bhl-he40^+ Icos^+ Il10^+*) (*Solé et al., 2023b*).

Further mono-omic analyses of the scRNAseq and scATACseq data of the scMultiome datasets provided additional information on the lineage relationships among the various T-cell sub-pools. Specifically, UMAP (Uniform Manifold Approximation and Projection) dimensional reduction of the scRNAseq data confirmed a high degree of transcriptional similarity between the TFH.1 cells from the KLH-induced TFH cell pool and the TFH-like cells contained within the BDC2.5mi/I-A$^{g7}$-NP-induced Tet+ pool (*Figure 1A*, middle). In fact, these two highly similar subsets only had five differentially expressed genes (|log2FC|>0.5 and adjusted p<0.05; *Actb, Ifi27l2a, Inpp4b, Nav2,* and *Tmsb10*) (*Figure 1—figure supplement 1*). Dimensional reduction of the scATACseq data showed that the open chromatin landscapes of the KLH-induced TFH.1 and TFH.2 sub-pools co-localized with those corresponding to the BDC2.5mi/I-A$^{g7}$-NP-induced Tet+ TFH and TR1 sub-pools (*Figure 1A*, right).

The transcriptional and epigenetic relationships among these various T-cell subsets were confirmed by hierarchical clustering of the two-dimensional scMultiome datasets (*Figure 1B*); despite coming from different mice and arising in response to different cues, the BDC2.5mi/I-A$^{g7}$-NP-induced TFH cells and the KLH-induced TFH.1/TFH.2 cells were more similar to each other than to other cell subsets within each sample.

## Extensive closure of open chromatin during the TFH-to-TR1 cell conversion within the Tet+ cell pool

Comparison of the scATACseq profiles of the KLH-induced TFH and BDC2.5mi/I-A$^{g7}$-NP-induced Tet+ TFH sub-pools indicated that TFH cells undergo massive closure of OCRs as they transdifferentiate into TR1-like and TR1 cells (*Figure 1C*). The data further indicated that most, albeit not all, of the OCRs that remain open in the terminally differentiated TR1 subset and, especially, the transitional TR1-like cells were already open at the TFH.1 cell stage (*Figure 1C*). Thus, the TFH-to-TR1 cell conversion process involves massive contraction of the chromatin and limited generation of new OCRs.

## Chromatin closure during the TFH-to-TR1 conversion within the Tet+ pool is associated with massive silencing of gene expression

Binding of TFs to their corresponding binding sites in DNA (TFBS) typically occurs in nucleosome-free regions in open chromatin. We next used the scMultiome datasets to investigate the effects of chromatin remodeling on gene expression during the TFH-to-TR1 cell conversion at the single-cell level. We focused on differentially OCRs in BDC2.5mi/I-A$^{g7}$-NP-induced TR1-like/TR1 cells (including both the TR1-like and TR1 sub-clusters) or TFH.1 cells (including both the BDC2.5mi/I-A$^{g7}$-NP-induced TFH and the KLH-induced TFH.1 cluster) as compared to their Tconv counterparts. We identified 688 genes that were associated with chromatin regions specifically open in TFH cells but not TR1-like/TR1 cells, and 545 genes that were associated with chromatin regions open in both TFH and TR1-like/TR1 cells or only TR1-like/TR1 cells.

Analyses of the scRNAseq data from the scMultiome dataset confirmed that chromatin closure during the TFH-to-TR1 cell conversion was accompanied by an equally extensive downregulation of gene expression. Specifically, there were 2086 genes that were differentially expressed in TR1 vs. TFH cells. Among these 2086 genes, 1820 (87.2%) were downregulated (e.g. *Cxcr5, Il4, Bcl6, Nfia, Stat4, Tcf7,* and *Tox2*) and only 266 (12.8%) were upregulated (e.g. *Bach2, Cd226, Ccr5, Ifng, Il10, Lef1, Prdm1,* and *Tbx21*) (*Figure 1D*).

Further analyses focusing on the genes that had closed chromatin regions during the TFH-to-TR1 conversion confirmed that there was a highly significant association between chromatin closure and downregulation of gene expression between TR1 and TFH cells. Specifically, 94% (n=217/231) of the genes associated with chromatin regions that had closed during the TFH-TR1 conversion, but only 31% (n=4/13) of the newly acquired OCRs, were downregulated (the remaining 69% were upregulated) (p=8.25e-14) (*Figure 1E*). Thus, massive chromatin closure during the TFH-to-TR1 conversion is accompanied by significant gene downregulation.

In contrast, the relative frequency of upregulated and downregulated genes among those linked to the 1245 regions equally accessible in TR1 and TFH cells (n=196) was similar to that seen for global gene expression changes during the TFH-to-TR1 conversion (86.7% genes downregulated) (p=0.92) (*Figure 1F*). Differential expression of these genes is thus likely regulated by factors other than chromatin exposure, such as TF availability, histone modifications, or DNA methylation.

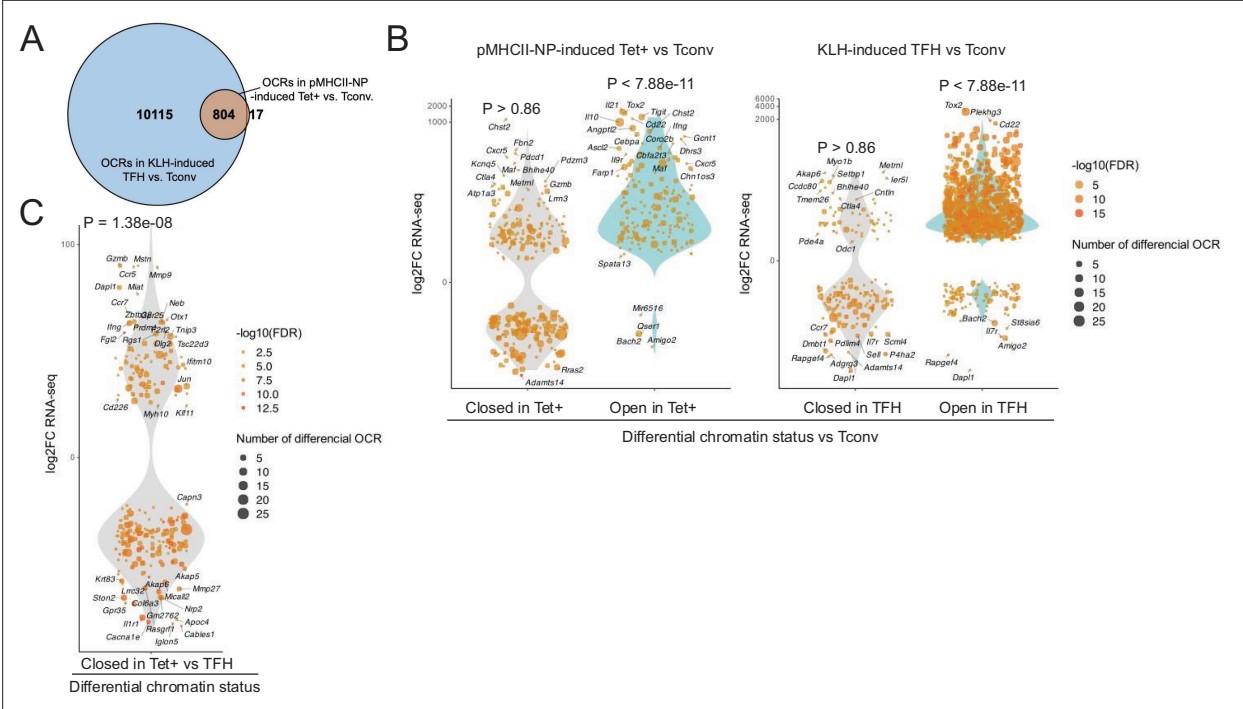

**Figure 2.** Chromatin accessibility in BDC2.5mi/I-A$^{g7}$-NP-induced Tet+ and KLH-DNP-induced T-follicular helper (TFH) cells vs. gene expression. (**A**) Euler's plot comparing the number and sharing of differential open chromatin regions (OCRs) between BDC2.5mi/I-A$^{g7}$-NP-induced Tet+ and KLH-induced TFH cells relative to their Tconv counterparts (n=4, 3, and 3, respectively) (adjusted p<0.01). (**B**) Associations between the number and status of differentially open or closed chromatin sites and gene expression as measured via bulk ATACseq (adjusted p<0.01) and RNAseq (adjusted p<0.01 and |log2FC|>2), respectively. Data correspond, from left to right, to BDC2.5mi/I-A$^{g7}$-NP-induced Tet+ vs. Tconv and KLH-induced TFH vs. Tconv. Each dot represents a gene, and its size is proportional to the number of associated OCRs. Dot color represents RNAseq differential analysis' -log10 of false discovery rate (FDR). Only genes with the highest log2FC value in each condition were labeled. (**C**) Associations between the status of chromatin accessibility in BDC2.5mi/I-A$^{g7}$-NP-induced Tet+ vs. KLH-induced TFH cells and gene expression as measured via bulk ATACseq (adjusted p<0.01) and RNAseq (adjusted p<0.01 and. |log2FC|>2), respectively. Each dot represents a gene, and its size is proportional to the number of associated OCRs. Dot color represents RNAseq differential analysis' -log10 of FDR. Only genes with the highest log2FC value in each condition were labeled. p-Values in (**B**) and (**C**) were calculated using Chi-square. KLH, keyhole limpet hemocyanin.

The online version of this article includes the following source data for figure 2:

**Source data 1.** Differential chromatin accessibility results by bulk ATAC-seq in pMHCII-NP-induced Tet+ cells and KLH-induced TFH cells, as compared to Tconv controls.

**Source data 2.** Number of differentially open chromatin regions (OCRs) and their associated genes in pMHCII-NP-induced Tet+ and KLH-induced TFH cells, compared to Tconv controls.

Collectively, the above data indicate that the TFH-to-TR1 cell conversion involves extensive remodeling of the chromatin and massive silencing of TFH gene expression.

## Contraction of the chromatin in pMHCII-NP-induced Tet+ vs. TFH cells at the bulk level

Bulk ATACseq studies of pMHCII-NP-induced Tet+ cells (~70% of which are TR1-like/TR1 cells; n=4) and KLH-induced TFH cells (~70% of which are TFH.1/TFH.2 cells; n=3) were consistent with the scMultiome data. KLH-induced TFH cells contained 13 times more differential OCRs (as compared to Tconv cells; n=3) than their BDC2.5mi/I-A$^{g7}$-NP-induced Tet+ counterparts (n=10,919 vs. 821, respectively). Furthermore, the overwhelming majority of the chromatin regions that are differentially exposed in pMHCII-NP-induced Tet+ cells (as compared to Tconv controls) (97.9%) are also differentially exposed in KLH-induced TFH cells (*Figure 2A* and *Figure 2—source data 1*). This includes genes such as *Batf*, *Bhlhe40*, *Cxcr5*, *Icos*, *Il10*, *Il21*, *Lag3*, *Maf*, *Nt5e*, *Pdcd1*, *Stat3*, *Tcf7*, and *Tox2*. In addition, the chromatin regions that are differentially open in the pMHCII-NP-induced Tet+ and/or KLH-induced TFH cell pools (as compared to Tconv cells, where these OCRs are closed) are significantly associated

with gene upregulation (p<7.88e-11); no such association is found for genes linked to closed chromatin in either pMHCII-NP-induced Tet+ or KLH-induced TFH cells (p=0.86) (*Figure 2B* and *Figure 2—source data 2*). Moreover, chromatin closure in BDC2.5mi/I-A$^{g7}$-NP-induced Tet+ cells relative to KLH-induced TFH cells was associated with downregulation of gene expression: 158 of the 341 genes that were downregulated in the former were linked to differentially closed OCRs (46.3%), as opposed to 94 of the 367 genes that were upregulated (25.6%) (p=1.38e-08) (*Figure 2C*).

Thus, studies of pMHCII-NP-induced Tet+ and KLH-induced TFH cells at the bulk level faithfully replicate the observations made using scMultiome. Together, they indicate that the TR1-like/TR1 cells contained within the Tet+ pool close a significant fraction of the chromatin as they transdifferentiate from TFH cells, leading to downregulation of gene expression, but the chromatin that remains open in TR1 cells is already exposed at the TFH cell stage. We acknowledge that, in the bulk ATACseq studies, the differences in the number of OCRs found in tetramer+ cells or KLH-induced TFH cells vs. naïve T-cells may be influenced by the intrinsic oligoclonality of the tetramer+ T-cell pool arising in response to repeated pMHCII-NP challenge (*Solé et al., 2023b*). However, we note that scATACseq studies of the tetramer+ T-cell pool found similar differences between the oligoclonal tetramer+ TFH sub-pool and its (also oligoclonal) tetramer+ TR1 counterparts (i.e. substantially higher number of OCRs in the former vs. the latter relative to naïve T-cells).

## H3K4me3, H3K27me3, and H3K27ac marks in genes upregulated during the TFH-to-TR1 cell conversion are already in place at the TFH cell stage

Histones can positively and negatively regulate gene expression upon undergoing post-translational modifications on N-terminal residues via acetylation, methylation, and ubiquitination of lysines; methylation and citrullination of arginines; or phosphorylation of serine, threonine, or tyrosine (*Greer and Shi, 2012*). Although some of these histone modifications are not involved in gene regulation but rather occur upon gene activation and RNA polymerase elongation, they are considered good epigenetic indicators of the status of the chromatin.

H3 is the histone that undergoes more epigenetic modifications. Acetylation of this histone, at K9, K14, K18, K23, or K27, is consistently associated with active transcription, by neutralizing the positive charge of lysine residues, weakening the H3-DNA interaction and enhancing accessibility of the chromatin to the transcription machinery. Generally, deposition of acetylated H3K27 (H3K27ac) is associated with gene expression and allows the identification of active/inactive or poised enhancers and active promoters (*Creyghton et al., 2010*). Unlike histone acetylation, histone methylation is electrically neutral and can be both activating or repressing, depending on the extent and lysine residue(s) involved. H3K4me3 deposition at transcriptional start sites (TSSs) is, like H3K27ac deposition, a marker of actively transcribed genes and is thought to imprint transcriptional 'memory' between generations (*Muramoto et al., 2010*). H3K27me3, generally found near CpG-rich promoters and intergenic regions, represses gene expression, even in the presence of H3K4me3, which prevents permanent silencing of the gene. The simultaneous presence of activating and repressing (bivalent) marks at the same location allows dynamic responsiveness to signals (*Jadhav et al., 2016*). Deposition of H3K27me3, like H3K4me3, has been linked to genomic imprinting through cell generations (*Raas et al., 2022*).

The scMultiome dataset described above indicated that there were 545 genes that were associated with regions of the chromatin that remain exposed as TFH.1 cells differentiate into TR1-like/TR1 cells and/or, to a much lesser extent, appear de novo in the latter. We therefore focused on this list of genes to investigate whether their lack of expression at the TFH cell stage was associated with absence of active/poised enhancers and promoters or with presence of repressive histone marks.

### H3K4me3

The KLH-induced TFH, pMHCII-NP-induced Tet+, and the Tconv subsets had a similar number of H3K4me3 marked regions/peaks (as defined via ChIPseq) (*Figure 3—source data 1*). As expected, most of these H3K4me3 marks (78%) were found at TSSs (*Figure 3—figure supplement 1*; *Figure 3—source data 1*). Representative chromosome track views are shown further below, in Figure 6.

The heatmaps shown in *Figure 3A* (left) show that the overall H3K4me3 deposition landscape in BDC2.5mi/I-A$^{g7}$-NP-induced Tet+ cells (containing ~70% TR1 cells) is closer to that seen in KLH-induced

**Figure 3.** Genome-wide distribution of H3K4me3, H3K27me3, and H3K27ac marks and active enhancers in BDC2.5mi/I-A$^{g7}$-NP-induced Tet+ cells and KLH-DNP-induced T-follicular helper (TFH) cells vs. differential gene expression. (**A**) Clustering heatmaps of all the regions enriched for H3K4me3 (left), H3K27me3 (middle), and H3K27Ac (right) deposition in BDC2.5mi/I-A$^{g7}$-NP-induced Tet+ cells, KLH-DNP-induced TFH cells, and Tconv cells. The intensity of the red color is proportional to the magnitude of enrichment vs. the corresponding background. (**B**) Bar plot comparing the relative percentages (X-axis) and absolute numbers (number annotations in each bar) of differentially marked regions for each histone modification (false discovery rate [FDR]<0.01). Top, middle, and bottom rows correspond to KLH-DNP-induced TFH vs. Tconv, BDC2.5mi/I-A$^{g7}$-NP-induced Tet+ vs. Tconv, and BDC2.5mi/I-A$^{g7}$-NP-induced Tet+ vs. KLH-DNP-induced TFH comparisons, respectively. KLH, keyhole limpet hemocyanin; NP, nanoparticle.

The online version of this article includes the following source data and figure supplement(s) for figure 3:

**Source data 1.** Detected peaks for either H3K27Ac, H3K4me3, and H3k27me3 histone depositions, in pMHCII-NP-induced Tet+, KLH-induced TFH cells, and Tconv cells.

**Source data 2.** Pairwise differential enrichment analysis of chromatin regions marked by H3K27Ac, H3K4me3, or H3K27me3 histone modifications, comparing pMHCII-NP-induced Tet+, KLH-induced TFH, and Tconv cells.

**Source data 3.** Categorization of differential histone deposition associated genes in pMHCII-NP-induced Tet+ and KLH-induced TFH cells, as compared to Tconv, for either H3K27Ac, H3K4me3, or H3K27me3 histone modifications.

**Figure supplement 1.** Differential gene deposition of H3K4me3, H3K27me3, and H3K27ac marks in BDC2.5mi/I-Ag7-NP-induced Tet+ cells, KLH-DNP-induced T-follicular helper (TFH) cells, and Tconv cells vs. differential gene expression.

TFH cells than in their Tconv counterparts (at the global level, including all chromatin regions, both open and closed at the TR1 cell stage) (**Figure 3—source data 1**). **Figure 3B** (left) shows the total number and relative percentage of differentially H3K4me3-marked regions (enriched for or depleted of H3K4me3) between KLH-induced TFH vs. Tconv, BDC2.5mi/I-A$^{g7}$-NP-induced Tet+ vs. Tconv and BDC2.5mi/I-A$^{g7}$-NP-induced Tet+ vs. KLH-induced TFH cells, respectively (**Figure 3—source data 2**). There were only 123 differentially marked H3K4me3 peaks between Tet+ and TFH cells (adjusted p<0.01) and, in the scMultiome dataset, most of these were linked to areas of the chromatin that were closed in TR1 cells as compared to their TFH precursors, except for two genes (*Ptpn11* and *Angptl2*), suggesting that differential H3K4me3 deposition at these genes is due to differential chromatin exposure. Further analysis of the data revealed that most of the H3K4me3 peaks found in genes linked to OCRs shared by TFH and TR1 cells at the single-cell level (77.6%; p<2.2e-16) were found in all three subsets (BDC2.5mi/I-A$^{g7}$-NP-induced Tet+, KLH-induced TFH cells, and Tconv cells), indicating that the corresponding genes are already marked for expression at both the naïve and TFH cell stages (**Figure 3—source data 3**).

Thus, the genes associated with regions of the chromatin that remain open in TR1 cells have nearly identical H3K4me3 deposition landscapes in both Tet+ and TFH cells.

## H3K27me3

We identified a total of 56,454 H3K27me3-marked peaks in the three subsets described above (**Figure 3—source data 1**). KLH-induced TFH cells had a significantly higher number of H3K27me3-marked peaks than Tconv or BDC2.5mi/I-A$^{g7}$-NP-induced Tet+ cells (42,274 vs. 10,944 and 3236, respectively) (p<2.2e-16). These H3K27me3 marks were found at the TSS (27.13%), or at intronic (19.29%) or intergenic regions (30%) (**Figure 3—figure supplement 1**) (**Figure 3—source data 1**).

The heatmaps shown in **Figure 3A** (middle; **Figure 3—source data 1**) show that as was the case for H3K4me3, the overall H3K27me3 deposition landscape in pMHCII-NP-induced Tet+ cells is closer to that seen in TFH cells than in their Tconv counterparts. **Figure 3B** (middle; **Figure 3—source data 2**) shows the number of differentially marked regions (enriched for or depleted of H3K27me3) between KLH-induced TFH vs. Tconv, BDC2.5mi/I-A$^{g7}$-NP-induced Tet+ vs. Tconv and BDC2.5mi/I-A$^{g7}$-NP-induced Tet+ vs. KLH-induced TFH cells. There were only 167 differentially marked regions between Tet+ and TFH cells (adjusted p<0.01), and most of these mapped to areas of the chromatin that are

closed during the TFH.1-to-TR1 transition, except for two genes (*Filip1l* and *Cdk8*). As was also the case for H3K4me3, most of the H3K27me3 marks found in genes associated with OCRs shared by both TFH and/or TR1 cells at the single-cell level (95.8%; p<2.2e-16) were shared by all three subsets (BDC2.5mi/I-A$^{g7}$-NP-induced Tet+, KLH-induced TFH cells, and Tconv cells), indicating that the corresponding genes already had this mark at the naïve and TFH cell stages (*Figure 3—source data 3*).

### H3K27Ac

As with H3K4me3, but unlike H3K27me3, the absolute number of H3K27Ac marks in each cell type were similar (~44,000/cell type). As expected, H3K27Ac marks were found at TSS (34%), intronic (24%), and intergenic locations (18%) (*Figure 3—figure supplement 1*) (*Figure 3—source data 1*).

The heatmaps shown in *Figure 3A* (right; *Figure 3—source data 1*) show that, as was the case for H3K4me3 and H3K27me3, the overall H3K27ac deposition landscape in BDC2.5mi/I-A$^{g7}$-NP-induced Tet+ cells is closer to that seen in KLH-induced TFH cells than in their Tconv counterparts. *Figure 3B* (right; *Figure 3—source data 2*) shows the number of differentially marked regions (enriched for or depleted of H3K27ac) between KLH-induced TFH vs. Tconv, BDC2.5mi/I-A$^{g7}$-NP-induced Tet+ vs. Tconv and BDC2.5mi/I-A$^{g7}$-NP-induced Tet+ vs. KLH-induced TFH cells. Remarkably, there were only seven regions that were differentially marked with H3K27Ac between Tet+ and TFH cells (adjusted p<0.01), in this case linked to *Cd247*, *Foxp1*, *Smco4*, and *Rab3ip*. Unlike the case for H3K4me3 and H3K27me3, most of the H3K27ac marks found in genes associated with OCRs shared by both TFH.1 and/or TR1 cells at the single-cell level (73.8%; p<2.2e-16) were shared by both BDC2.5mi/I-A$^{g7}$-NP-induced Tet+ and KLH-induced TFH but not Tconv cells (*Figure 3—source data 3*), indicating that these genes were marked with H3K27ac at the TFH cell stage.

We note that, although in the representative chromosome track views shown in Figure 6C, there appear to be differences in the intensity of the peaks, thorough statistical analyses involving signal background for each condition and p-value adjustment did not support differential enrichment for histone deposition around the *Il10* gene between pMHCII-NP-induced tetramer+ T-cells and KLH-induced TFH cells.

Collectively, these results suggest that most of the genes that are upregulated during the TFH-to-TR1 cell conversion have H3K4me3, H3K27me3, and H3K27ac marks that are already in place at the TFH stage.

## The methylation status of most of the genes that remain accessible at the TR1 cell stage is already imprinted at the TFH stage

DNA methylation is based on the covalent binding of a methyl group to the C-5 position of a cytosine ring of DNA. In adult mammalian cells, 98% of DNA methylation targets cytosines in CpG dinucleotides, is directly associated with transcriptional silencing, and is maintained through cell division (i.e. is a heritable epigenetic trait) (*Schübeler, 2015*). Demethylation is an ultimate defining step of cell identity and is associated with long-term enhancer accessibility (*Barnett et al., 2020*). Consequently, differentiated cells possess a stable and unique methylome structure that regulates their cell-specific transcriptomic profile.

To investigate the potential contribution of changes in DNA methylation to gene expression as BDC2.5mi/I-A$^{g7}$-NP-induced TFH.1 cells transdifferentiate into TR1 cells, we performed genome-wide bisulfite sequencing of BDC2.5mi/I-A$^{g7}$-NP-induced Tet+, KLH-induced TFH cells, and Tconv cells. We focused our analysis on differentially methylated regions (DMRs; q-value<0.05) (*Figure 4—source data 1*). Among all the DMRs found in BDC2.5mi/I-A$^{g7}$-NP-induced Tet+ and KLH-induced TFH cells vs. Tconv cells, 43.2% were shared by BDC2.5mi/I-A$^{g7}$-NP-induced Tet+ and KLH-induced TFH cells, 33.9% were unique to KLH-induced TFH cells and 24.8% were unique to BDC2.5mi/I-A$^{g7}$-NP-induced Tet+ cells (*Figure 4A* and *Figure 4—source data 1*).

Most of these DMRs were primarily found in intronic CpG islands, followed by exonic and then intergenic regions and gene promoters (*Figure 4—figure supplement 1* and *Figure 4—source data 1*). The differentially hypomethylated regions found in BDC2.5mi/I-A$^{g7}$-NP-induced Tet+ cells relative to Tconv cells were associated with 2714 genes. Most of these genes (70%; p<2.2e-16) were also differentially hypomethylated in KLH-induced TFH cells (*Figure 4B*, left; *Figure 4—source data 1 and 2*). Likewise, of the 2748 genes that were associated with differentially hypermethylated regions in BDC2.5mi/I-A$^{g7}$-NP-induced Tet+ cells, most (71%; p<2.2e-16) also harbored differentially

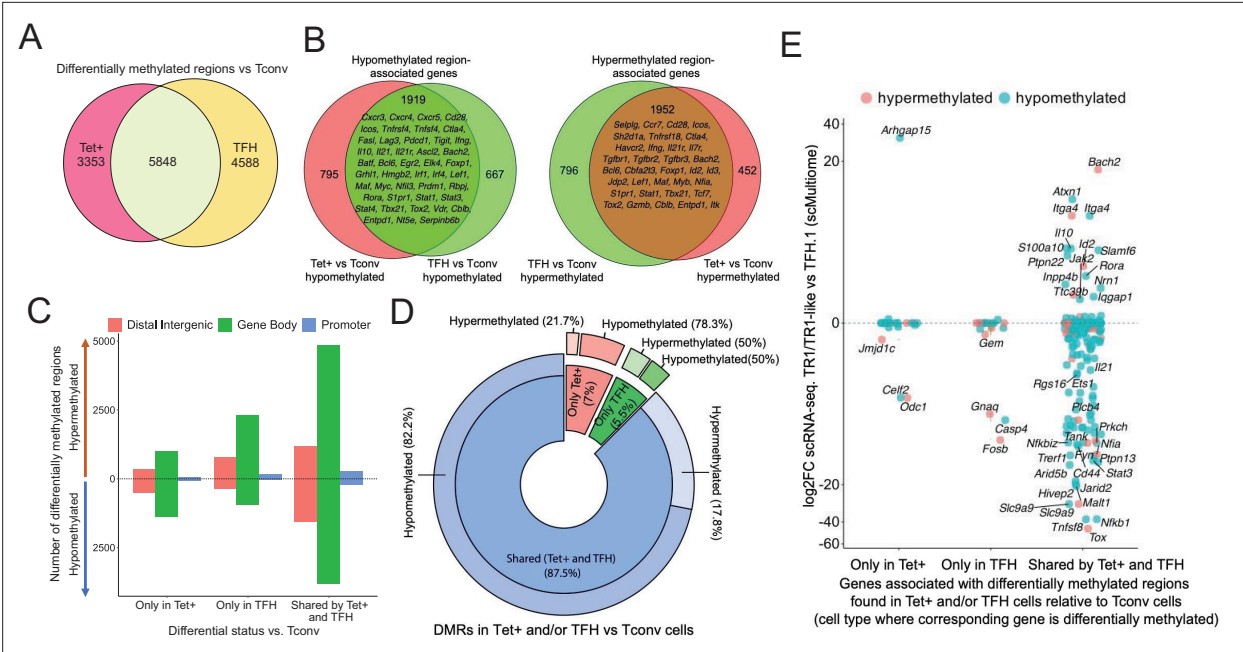

**Figure 4.** Gene methylation status in BDC2.5mi/I-A$^{g7}$-NP-induced Tet+, KLH-DNP-induced T-follicular helper (TFH), and Tconv cells. (**A**) Venn diagram of differentially methylated regions (DMRs) in BDC2.5mi/I-A$^{g7}$-NP-induced Tet+ and KLH-DNP-induced TFH vs. Tconv cells. (**B**) Venn diagrams of differentially hypomethylated (left) or hypermethylated (right) regions shared by BDC2.5mi/I-A$^{g7}$-NP-induced Tet+ cells and KLH-DNP-induced TFH cells as compared to their Tconv counterparts. Gene names found in the 106 TFH/TR1/Treg gene list in ***Supplementary file 1***, with shared methylation status, are indicated. (**C**) Bar plot showing the numbers of differentially hypo- or hypermethylated regions (separated by gene region: promoter, gene body, or distal intergenic) in BDC2.5mi/I-A$^{g7}$-NP-induced Tet+ and KLH-DNP-induced TFH vs. Tconv cells, respectively. DMRs are classified, from left to right, into those only found in BDC2.5mi/I-A$^{g7}$-NP-induced Tet+ cells (only Tet), KLH-DNP-induced TFH cells (only TFH), or found in both subsets (shared by tet+ and TFH). (**D**) Pie-donut chart showing the distribution of DMRs (hyper- and hypomethylated status) in genes associated with open chromatin regions (OCRs) (n=328) shared by TFH.1 and TR1/TR1-like cells (identified via single-cell Multiome [scMultiome]). (**E**) Jitter plot comparing differential gene expression between TR1/TR1-like and TFH.1 cells (as determined by scMultiome) and differential methylation status, focusing on the genes that remain open at the TR1/TR1-like cell stage, as determined by scMultiome. DMR-associated genes are classified based on their cell pool specificity, i.e., only found in BDC2.5mi/I-A$^{g7}$-NP-induced Tet+ cells (only Tet+), KLH-DNP-induced TFH cells (only TFH), or both (shared by Tet+ and TFH). Color depicts the methylation status (hypo- or hypermethylated) of the regions associated with these genes. No significant correlation between methylation status and differential gene expression was found. KLH, keyhole limpet hemocyanin; NP, nanoparticle.

The online version of this article includes the following source data and figure supplement(s) for figure 4:

**Source data 1.** Pairwise differentially methylated regions (DMR) comparing pMHCII-NP-induced Tet+, KLH-induced TFH, and Tconv cells.

**Source data 2.** Relative methylation status, chromatin annotation, relevant associated genes, and number of regions differentially methylated between pairwise comparison of pMHCII-NP-induced Tet+, KLH-induced TFH, and Tconv cells.

**Source data 3.** Differential methylation status of genes associated with chromatin regions accessible in TFH.1 and TR1 cells as determined by scATAC-seq.

**Source data 4.** List of genes associated with chromatin regions remaining open during TFH to TR1 conversion with corresponding data on differential expression, differential methylation, chromatin accessibility, and histone deposition.

**Figure supplement 1.** Differences in gene region distribution of differentially methylated regions (DMRs) among BDC2.5mi/I-A$^{g7}$-NP-induced Tet+, KLH-DNP-induced T-follicular helper (TFH), and Tconv cells.

**Figure supplement 2.** Differential DNA methylation at or near the *Il2*, *Il10*, and *Il21* loci between BDC2.5mi/I-A$^{g7}$-NP-induced Tet+ or KLH-DNP-induced T-follicular helper (TFH) and Tconv cells.

**Figure supplement 3.** Relative contribution of different epigenetic marks to changes in gene expression during the TFH-to-TR1 cell conversion (TFH vs. Tconv).

**Figure supplement 4.** Relative contribution of different epigenetic marks to changes in gene expression during the TFH-to-TR1 cell conversion (Tet+ vs. Tconv).

**Figure supplement 5.** Relative contribution of different epigenetic marks to changes in gene expression during the TFH-to-TR1 cell conversion (Tet+ vs. TFH).

hypermethylated regions in KLH-induced TFH cells (*Figure 4B*, right; *Figure 4—source data 1 and 2*). *Figure 4C* provides a graphical representation of the genic location of these DMRs as a function of whether they are shared between Tet+ and TFH vs. Tconv cells and their methylation status. Thus, BDC2.5mi/I-A$^{g7}$-NP-induced Tet+ cells and KLH-induced TFH cells share a remarkably similar methylome (*Figure 4—source data 1*).

We next focused our attention on genes whose chromatin was accessible in both TFH.1 and TR1 cells as determined by scATACseq (i.e. excluding genes silenced by chromatin closure). We classified these genes into three groups: (1) carrying DMRs in BDC2.5mi/I-A$^{g7}$-NP-induced Tet+ but not KLH-induced TFH cells; (2) carrying DMRs in KLH-induced TFH cells but not BDC2.5mi/I-A$^{g7}$-NP-induced Tet+ cells; and (3) shared by both populations. Notably, 328 of 545 genes associated with accessible chromatin in both TFH and TR1-like/TR1 cells at the single-cell level harbor DMRs in BDC2.5mi/I--A$^{g7}$-NP-induced Tet+ and/or KLH-induced TFH cells vs. Tconv cells, and most of these genes (87.5%; p<1.85e-5) have a similar methylation status in both BDC2.5mi/I-A$^{g7}$-NP-induced Tet+ and KLH-induced TFH cells (n=236 hypomethylated; n=51 hypermethylated) (*Figure 4D* and *Figure 4—source data 3*). Thus, the genes that remain accessible during the TFH-to-TR1 cell differentiation process share an even greater degree of DNA methylation status than when considering all genes regardless of chromatin accessibility.

Of the 328 accessible genes associated with DMRs in BDC2.5mi/I-A$^{g7}$-NP-induced Tet+ and/or KLH-induced TFH cells vs. Tconv cells, 159 (48.5%) were differentially expressed between TR1 and TFH cells as determined by scRNAseq. As expected, based on the data shown above, there was no correlation between methylation status and differential gene expression (*Figure 4E* and *Figure 4—source data 3*) (p=0.92). Although most of the accessible genes sharing their methylation status in both BDC2.5mi/I-A$^{g7}$-NP-induced Tet+ and KLH-induced TFH cells were not differentially expressed in TR1 vs. TFH.1 cells (50.2%), 41.1% were downregulated (e.g. *Cxcr5*, *Il21*, *Pdcd1*, *Ctla4*, *Tigit*, *Maf*, *Nfia*, and *Tox2*) and 8.7% were upregulated (e.g. *Il10*, *Bach2*, and *Tbx21*).

This apparent inheritance of gene methylation status by pMHCII-induced TR1 cells from their TFH precursors is further illustrated by direct comparison of the methylation status of TFH and TR1-specific genes, such as *Il2* and *Il10*, or genes expressed by both, such as *Il21*. *Il2* is highly expressed in TFH but not TR1 cells and yet it is differentially hypomethylated in both BDC2.5mi/I-A$^{g7}$-NP-induced Tet+ and KLH-induced TFH cells as compared to Tconv cells (*Figure 4—figure supplement 2*). Likewise, *Il10*, which is expressed by TR1 cells but not TFH cells is already significantly hypomethylated in several regions upstream of the TSS in KLH-induced TFH cells (and BDC2.5mi/I-A$^{g7}$-NP-induced Tet+ cells) as compared to their Tconv counterparts; in fact, when these two cell subsets are compared directly to each other, only relatively minor differences in the methylation status of *Il2* and *Il10* can be seen (*Figure 4—figure supplement 2*). As expected, *Il21*, expressed by both TFH and TR1 cells, is also hypomethylated in both T-cell subsets as compared to Tconv cells (*Figure 4—figure supplement 2*).

Collectively, the above data suggest that the methylation status of most of the genes that remain accessible at the TR1 cell stage is already imprinted at the TFH stage, and that most gene expression differences between TR1 and TFH cells cannot be accounted for changes in the overall methylation status of the corresponding genes. In a small number of cases, however, the TFH-to-TR1 cell conversion is accompanied by further gene demethylation of TR1-specific genes (i.e. *Il10*) and remethylation of TFH-specific ones (i.e. *Cxcr5*).

## Changes in gene expression during the TFH-to-TR1 cell conversion are largely dissociated from redistribution of epigenetic marks

To further define the contribution of the various epigenetic modifications discussed above on gene expression, we again focused on the genes associated with chromatin regions that remain open as TFH cells become TR1. We then ranked these genes according to gene expression changes, from upregulated to downregulated in TFH or Tet+ cells as compared to Tconv cells (*Figure 4—figure supplements 3–4*) or in Tet+ vs. TFH cells (*Figure 4—figure supplement 5*; *Figure 4—source data 4*). Whereas for most genes, differential gene expression in TFH or Tet+ vs. Tconv cells is associated with differential gene methylation, open chromatin, and H3K27ac and/or H3K4me3 deposition, very few differences were noted in all these readouts when comparing Tet+ to TFH cells.

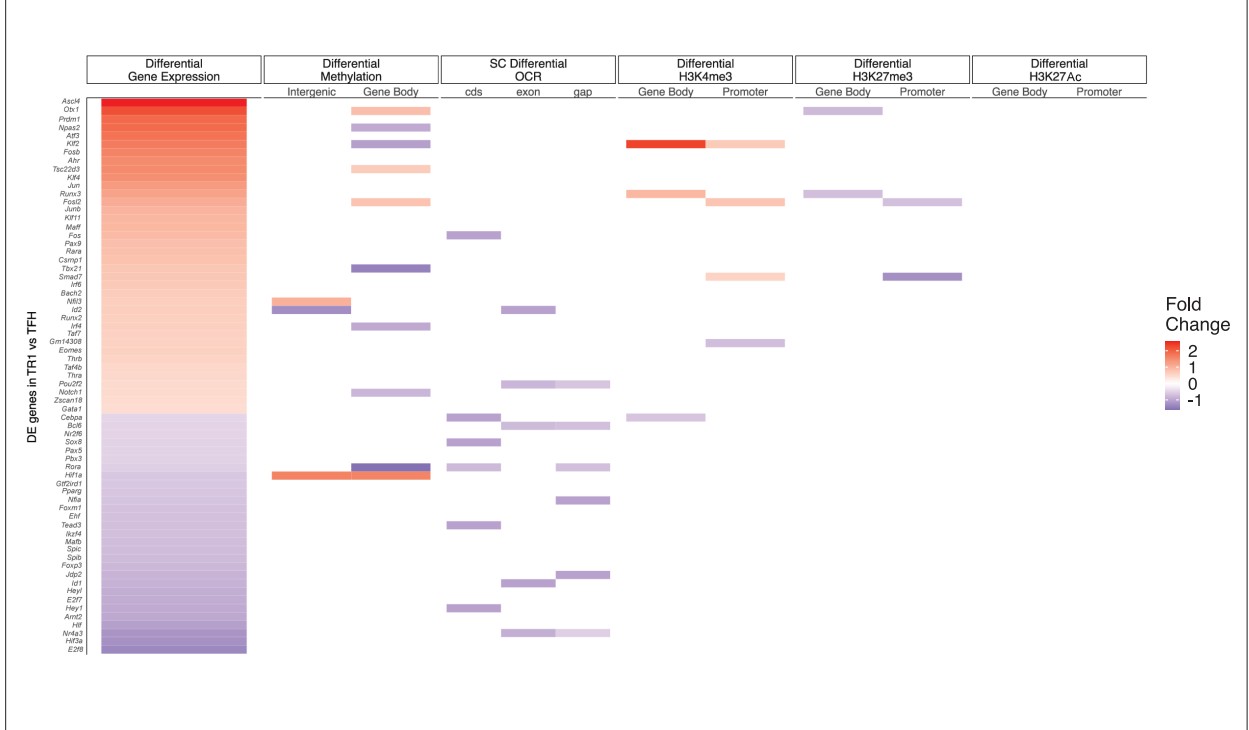

**Figure 5.** Relative contribution of different epigenetic marks to changes in the expression of differentially expressed transcription factor (TF)-coding genes during the TFH-to-TR1 cell conversion. Heatmap depicting the presence of different epigenetic marks (from left to right: differential methylation, differential open chromatin regions [OCRs], differential H3K27ac deposition, differential H3K4me3 deposition, and differential H3K27me3 deposition) in BDC2.5mi/I-A$^{g7}$-NP-induced Tet+ vs. KLH-DNP-induced TFH cells. Data correspond to differentially expressed TF-coding genes between TR1 and TFH.1 cells as determined by the single-cell Multiome (scMultiome) analyses. Differential epigenetic data is scaled for each technique and when multiple genomic regions are associated to a gene, the average is provided. Genes are arranged from most to least upregulated, followed by least to most downregulated (fold change). No differentially enriched sites for H3K27Ac histone deposition were associated with differential expression of these genes. KLH, keyhole limpet hemocyanin; TFH, T-follicular helper; TR1, T-regulatory type 1; NP, nanoparticle.

The online version of this article includes the following source data for figure 5:

**Source data 1.** Comparison of transcription factor-coding genes that are upregulated or downregulated during the TFH-to-TR1 conversion, detailing chromatin accessibility changes (OCR closure) and associated epigenetic marks for each gene.

## Loss of TFH-specific TF gene expression during the TFH-to-TR1 conversion is associated with chromatin closure

The above data collectively suggest that transdifferentiation of TFH cells into TR1 cells is driven by changes in the expression of TFH-stabilizing and TR1-promoting TFs. This, coupled to the extensive closure of chromatin sites in TFH cells as they become TR1 cells, suggested that changes in TF expression, particularly the loss of TFH-associated TFs, might be driven, in part, via chromatin remodeling of the coding loci. To investigate this, we compared the types and direction (expression-promoting or suppressing) of the various epigenetic modifications studied above on TF-coding genes as a function of upregulation or downregulation. As shown in *Figure 5*; *Figure 5—source data 1*, the TF-coding genes that are downregulated during the TFH-to-TR1 conversion, unlike those that are upregulated (based on the scMultiome data), close a significant number of OCRs (13/29 of downregulated TF-coding genes had closed OCRs, as compared to only 4/38 of upregulated TF-coding genes, p<0.0001). As expected, based on the epigenetic similarity of TFH vs. TR1 cells, upregulation of TF-coding genes was largely dissociated from the epigenetic marks studied here.

## TR1 cells inherit active enhancers from their TFH precursors

Gene expression is driven by the sequential recruitment of DNA-binding TFs (bound to proximal promoters and/or distal GREs), non-DNA-binding cofactors and the transcription machinery to the core promoter. Enhancers are GREs that positively activate transcription in primed gene promoters

found thousands of kb away and even on different chromosomes but are proximal in the three-dimensional structure of the chromatin in the nucleus. Whereas active enhancers are typically marked with H3K4me1, H3K27ac, and some H3K4me3, poised enhancers contain both H3K4me1 and the repressive H3K27me3 mark (*Creyghton et al., 2010*; *Heintzman et al., 2007*; *Rada-Iglesias et al., 2011*; *Pekowska et al., 2011*). Active enhancers target genes marked with H3K27ac and H3K4me1 at and downstream of their TSS, respectively (*Creyghton et al., 2010*; *Heintzman et al., 2007*; *Rada-Iglesias et al., 2011*; *Pekowska et al., 2011*).

Thus, whereas ATACseq helps to identify areas of open chromatin associated with various regulatory elements such as enhancers, silencers, and promoters, H3K27Ac ChIPseq helps locate class I

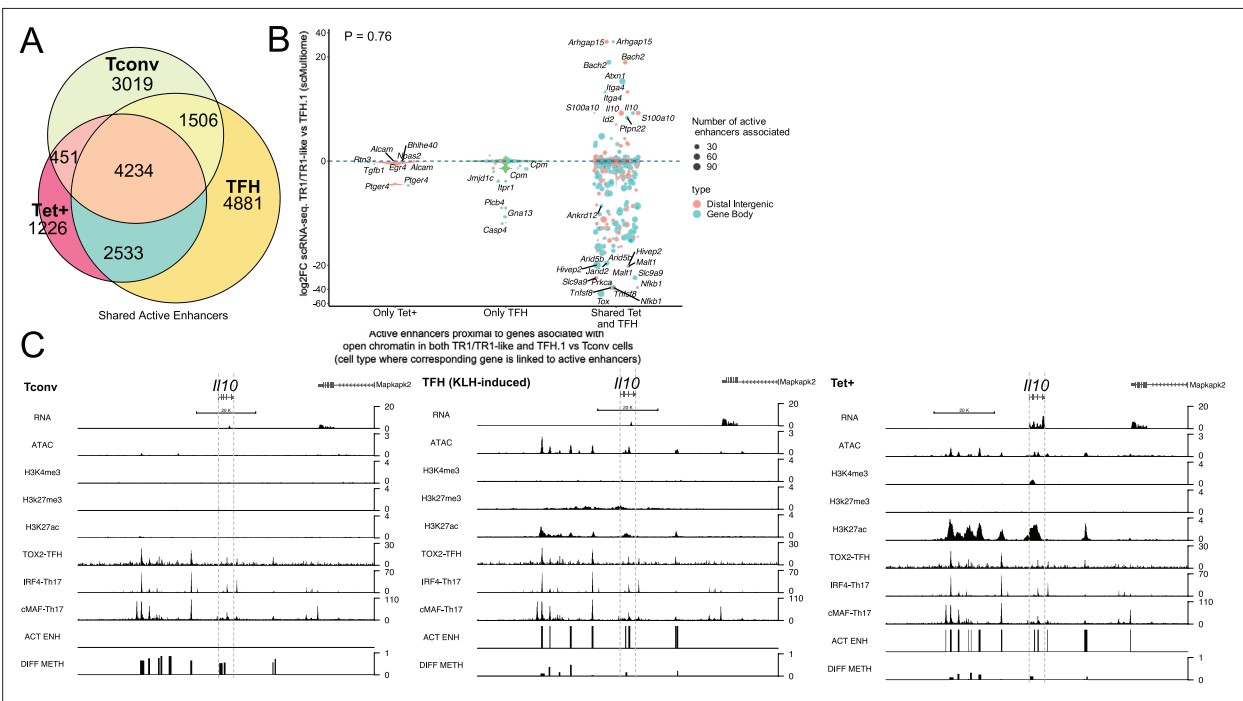

**Figure 6.** T-regulatory type 1 (TR1) cells inherit hypomethylated active enhancers from their T-follicular helper (TFH) precursors. (**A**) Euler's plot showing total active enhancer sharing by BDC2.5mi/I-A$^{g7}$-NP-induced Tet+ cells, KLH-DNP-induced TFH cells, and Tconv cells. Overlapping of active enhancers in BDC2.5mi/I- A$^{g7}$-NP-induced Tet+ cells and KLH-DNP-induced TFH cells was significantly higher than overlapping between BDC2.5mi/I-A$^{g7}$-NP-induced Tet+ and Tconv cells (Pearson's Chi-squared test with Yates' continuity correction p<2.2e-16). (**B**) Jitter plot comparing differential gene expression between TR1/TR1-like and TFH.1 cells (as determined by single-cell Multiome [scMultiome]), focusing on the genes that remain open at the TR1/TR1-like cell stage, and both distribution and number of active enhancers per gene as a function of their cell pool specificity (i.e. only found in BDC2.5mi/I- A$^{g7}$-NP-induced Tet+ cells (only Tet+), KLH-DNP-induced TFH cells (only TFH), or both (shared by Tet+ and TFH)). Color and size depict the region type (gene body or intergenic) and the number of active enhancers per gene, respectively. Gene names are displayed for all the genes except when more than 20 dots are in the same region of the plot. No significant correlation between active enhancer distribution and differential gene expression (Wilcox test for differential analysis: adjusted p<0.05) was found using Pearson's Chi-square test, p=0.76. (**C**) Figure displays genome tracks for the various readouts examined herein in Tconv CD4+ T-cells (left), KLH-DNP-induced TFH cells (middle), and BDC2.5mi/I-A$^{g7}$-NP-induced Tet+ cells (right). Tracks correspond, from top to bottom, to reads for RNAseq (n=4), ATACseq (n=3), ChIPseq (n=1) for H3K4me3, H3K27me3, and H3K4me3 immunoprecipitation, respectively; ChIPseq (n=1) for Tox- 2 (TFH cells), Irf4 (Th17 cells), and cMaf (Th17 cells) deposition, respectively (see main text for references), active enhancers and differential methylation, respectively. Visualization reads were normalized to total sequencing depth per sample using BPM (bins per million) and, where replicates were available, height mean per bin was also computed. ChIPseq data was also normalized for input (non-immunoprecipitated) sequenced reads. Height (y-axis) is equivalent to the normalized number of mapped reads in each region. Active enhancers (ACT ENH) were predicted as overlapping regions of open chromatin region (OCR) (ATACseq) and H3K27Ac deposited peaks (ChIPseq H3K27Ac) and are depicted as absent or present in each region. DIFF METH shows differentially methylated regions (n=3) obtained comparing BDC2.5mi/I-A$^{g7}$-NP-induced Tet+ cells and KLH-DNP-induced TFH to Tconv cells. Height corresponds to the relative mean methylation value for each region. KLH, keyhole limpet hemocyanin; NP, nanoparticle.

The online version of this article includes the following source data for figure 6:

**Source data 1.** Number of total and shared mapped active enhancers between pMHCII-NP-induced Tet+, KLH-induced TFH, and Tconv cells.

**Source data 2.** Differential gene expression by scRNA-seq between TR1 and TFH cells of genes associated to active enhancers overlapping accessible chromatin regions in both TFH and TR1 cells.

active enhancer and promoter elements. To map the location of active enhancers in BDC2.5mi/I-A$^{g7}$-NP-induced Tet+ cells and KLH-induced TFH cells, we carried out an integrated analysis of both data-sets (areas of open chromatin and H3K27ac deposition) in both cell pools. OCRs containing H3K27Ac peaks, excluding those located within 2 kb of the TSS (i.e. overlapping promoters), were considered to represent active enhancers. As expected, based on the data presented above, the BDC2.5mi/I--A$^{g7}$-NP-induced Tet+ pool shared significantly more active enhancers with KLH-induced TFH cells (n=6767/8444; 80.2%) than with Tconv cells (n=4685/8444; 55.5%) (p<2.2e-16) (*Figure 6A* and *Figure 6—source data 1*).

We next focused on active enhancers proximal to genes linked to accessible chromatin in both TFH and TR1 cells, as defined via scATACseq. We divided the corresponding active enhancers into three sub-groups: (1) those exclusively found in BDC2.5mi/I-A$^{g7}$-NP-induced Tet+ cells; (2) only found in KLH-induced TFH cells, and (3) shared by both cell types. As shown in *Figure 6B*, most of the genes that remain open as the cells transition from the TFH state to its TR1 counterparts (i.e. are not closed), already harbor active enhancers in TFH cells. Specifically, most of the 396 genes that are associated with accessible chromatin in both TR1-like/TR1 and TFH cells and are marked with active enhancers (89.6%; p<2.2e-16) display such enhancers in both the BDC2.5mi/I-A$^{g7}$-NP-induced Tet+ and KLH-induced TFH pools.

We then investigated whether differences in the expression of these genes, as defined via scRNAseq, were associated with differences in the number and/or location of active enhancers. As expected, given the high epigenetic poised state for genes upregulated at the TR1 cell stage, at the precursor (TFH) cell stage for all readouts examined so far, there was no significant correlation between active enhancer distribution and differential gene expression (*Figure 6B* and *Figure 6—source data 2*) (p=0.76). In fact, as also noted for genes associated with chromatin regions that remained open at the TR1 cell stage, which were mostly downregulated (*Figure 1F*), most genes marked with active enhancers at the TFH stage (68.9%; p<2.2e-16) were downregulated in TR1 cells (e.g. *Cxcr5*, *Il21*, *Pdcd1*, *Tigit*, *Egr2*, *Maf*, *Nfia*, and *Tox2* to name a few).

Together, these data suggest that: (1) most of the genes that remain open as BDC2.5mi/I-A$^{g7}$-NP-induced TFH cells transition into TR1-like/TR1 cells also share active enhancers in both subsets; and (2) differences in the expression of these genes are likely mediated by other factors, such as DNA demethylation and/or differential TF availability.

## Most of the upregulated genes at the TR1 stage had already demethylated their distal GREs at the TFH stage

Detailed analyses of the intergenic DMRs found in KLH-induced TFH cells (and shared with BDC2.5mi/I-A$^{g7}$-NP-induced Tet+ cells) revealed a striking overlap with active enhancers. *Figure 6C* illustrates the location of DMRs around *Il10* locus relative to the various transcriptional and epigenetic readouts explored herein, including active enhancers, in Tconv, BDC2.5mi/I-A$^{g7}$-NP-induced Tet+ and KLH-induced TFH cells. Several lines of evidence suggest that the patterned hypomethylation status of distal gene regulatory elements of TR1 genes in TFH precursors define the TR1-poised nature of the TFH epigenome. First, DNA methylation is generally not permissive for transcription (*Schübeler, 2015*), even at active enhancers. Second, enhancer demethylation is highly cell type-specific and accurately predicts target gene transcription (*Schlesinger et al., 2013*). Third, differential methylation among cell types is greatest at distal gene regulatory elements than in promoters (*Schlesinger et al., 2013*; *Stadler et al., 2011*). Finally, demethylation at these sites appears to be a required final step in enhancer activation during cell fate transitions, leading to the stabilization of cell line identity (*Barnett et al., 2020*).

To reveal the identity of differentially expressed genes proximal to 'distal' GREs marked by open chromatin, H3K27ac deposition and hypomethylation (excluding promoters), we interrogated our active enhancer dataset for Tet+ and TFH-specific DMRs (vs. Tconv cells). The vast majority of DMRs mapping to active enhancers found in BDC2.5mi/I-A$^{g7}$-NP-induced Tet+ cells (92.5%; p<2.2e-16) were hypomethylated (including *Icos*, *Ctla4*, *Pdcd1*, *Tigit*, *Il10*, *Irf4*, *Maf*, and *Prdm1*) (bottom pie chart in *Figure 7A*; *Figure 7—source data 1*). This was also true for KLH-induced TFH cells (n=1051/1185; 88.7%; p<2.2e-16) (e.g. *Cxcr5*, *Il10*, *Il21*, *Bach2*, *Nfil3*, *Nfil3*, and *Tox2*) (middle pie chart in *Figure 7A*; *Figure 7—source data 1*). In fact, a large fraction of the differentially methylated active enhancers found in these subsets were shared between BDC2.5mi/I-A$^{g7}$-NP-induced Tet+ and KLH-induced TFH

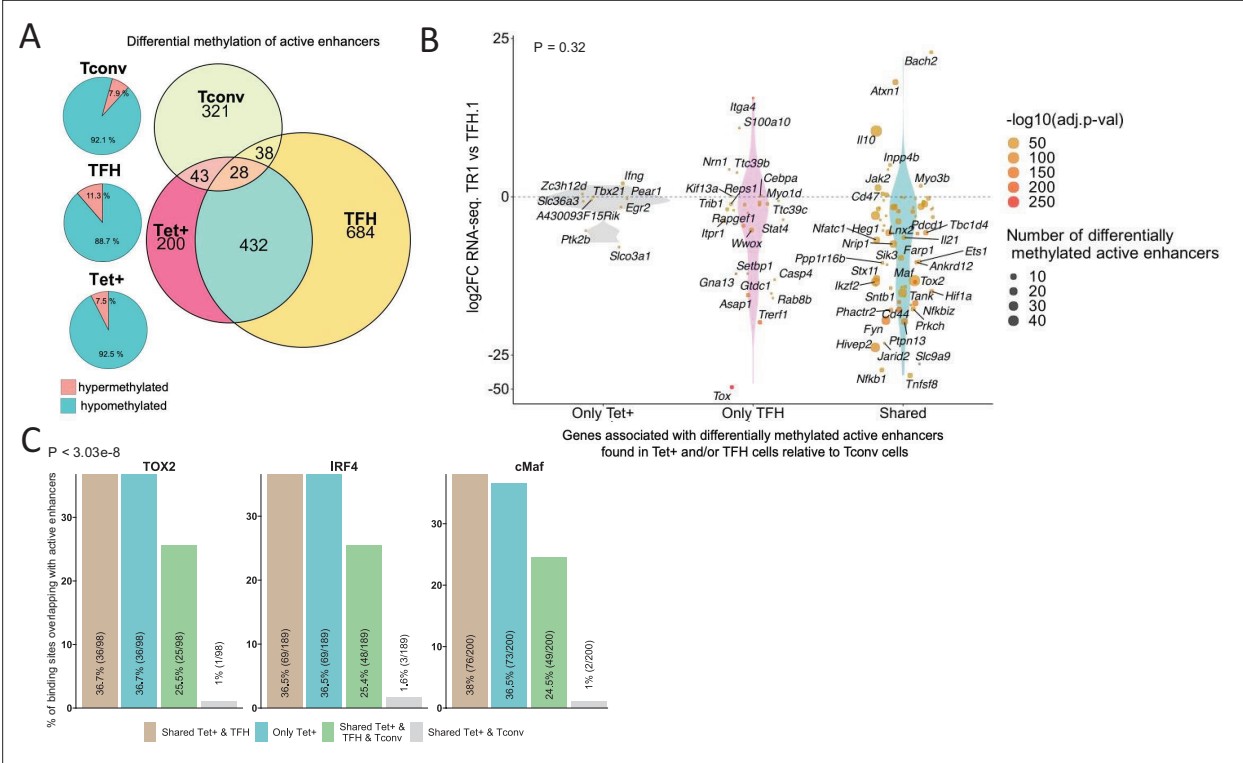

**Figure 7.** Extensive sharing of differentially methylated active enhancers carrying Tox2, IRF4, and/or c-MAF-binding sites in BDC2.5mi/I-A$^{g7}$-NP-induced Tet+ and KLH-DNP-induced T-follicular helper (TFH) cells vs. Tconv cells. (**A**) Euler's plots with differentially methylated active enhancers for each comparison: BDC2.5mi/I- A$^{g7}$-NP-induced Tet+ cells vs. Tconv (Tet+; red), KLH-DNP-induced TFH vs. Tconv (TFH; orange), and Tconv vs. BDC2.5mi/I-A$^{g7}$-NP-induced Tet+ cells (Tconv; green). Pie charts on the left show the methylation status (hypomethylated or hypermethylated) of each subset of active enhancers for the same subset-to-subset comparisons. The BDC2.5mi/I-A$^{g7}$-NP-induced Tet+ and KLH-DNP-induced TFH cell subsets share significantly more differentially methylated active enhancers among each other than with Tconv cells (Pearson's Chi-squared test with Yates' continuity correction; p<2.2e-16). (**B**) Jitter plot of differentially methylated active enhancers in BDC2.5mi/I- A$^{g7}$-NP-induced Tet+ cells (Tet+) and KLH-DNP-induced TFH (TFH) vs. Tconv cells vs. differential gene expression along the TFH.1-TR1 axis as determined by single-cell Multiome (scMultiome). Plot corresponds to genes that remain accessible as TFH.1 cells transdifferentiate into TR1/TR1-like cells. The differentially methylated enhancers linked to these genes were classified as being specific for BDC2.5mi/I-A$^{g7}$-NP-induced Tet+ cells (only Tet+), exclusive for KLH-DNP-induced TFH cells (only TFH) or shared by both Tet+ and TFH cells. Gene bubble color depicts -log10 of adjusted p-value from scRNAseq differential analysis, and bubble size depicts the number of regions associated per gene. Gene names are displayed for all the genes except when more than 20 dots are in the same region of the plot. No statistically significant associations between the presence of differentially hypo- or hypermethylated active enhancers and gene expression differences were found (Chi-square test: p=0.32). (**C**) Overlap of active enhancers with Tox2, IRF4, and c-MAF binding sites. Histogram plot compares the relative proportion of the active enhancers linked to genes that are accessible in both TR1/TR1-like and TFH.1 cells (as defined via scMultiome) and that are upregulated in TR1/TR1-like vs. TFH.1 cells (as determined via scMultiome), which overlap with binding sites for Tox-2 in TFH cells, and IRF4 or c-MAF in Th17 cells. Bars correspond from left to right, to: (1) binding sites shared by both BDC2.5mi/I-A$^{g7}$-NP-induced Tet+ and KLH-DNP-induced TFH cells (shared Tet+ & TFH); (2) exclusively found in the BDC2.5mi/I- A$^{g7}$-NP-induced Tet+ pool (only Tet+); (3) shared by BDC2.5mi/I-A$^{g7}$-NP-induced Tet+, KLH-DNP-induced TFH and Tconv cells (shared Tet+ & TFH & Tconv); and (4) shared by BDC2.5mi/I-A$^{g7}$-NP-induced Tet+ and Tconv cells (shared Tet+ & Tconv). KLH, keyhole limpet hemocyanin; TR1, T-regulatory type 1; NP, nanoparticle.

The online version of this article includes the following source data for figure 7:

**Source data 1.** Methylation status of mapped active enhancers in pMHCII-NP-induced Tet+ and KLH-induced TFH cells.

**Source data 2.** Overlapping of TOX-2, IRF4, and c-MAF binding sites with active enhancers sharing open chromatin between TR1 and TFH and associated to genes upregulated at the TR1 cell stage.

cells (65% and 35%, respectively; Euler's plot in *Figure 7A*; *Figure 7—source data 1*). In contrast, only 10% and 10.71% of differentially methylated active enhancers found in BDC2.5mi/I-A$^{g7}$-NP-induced Tet+ cells and KLH-induced TFH cells, respectively, were shared with Tconv cells (Euler's plot in *Figure 7A*) (p=0.02) (*Figure 7—source data 1*). In addition, most of the DMRs overlapping active enhancers found in TFH and Tet+ cells (92.1%) are hypomethylated (top pie chart in *Figure 7A*; *Figure 7—source data 1*).

We next explored the methylation status of active enhancers linked to genes that remain accessible as TFH cells transdifferentiate into TR1 cells. The differentially methylated enhancers linked to these genes were classified as: (1) specific for BDC2.5mi/I-A$^{g7}$-NP-induced Tet+ cells; (2) specific for KLH-induced TFH cells; and (3) shared by both BDC2.5mi/I-A$^{g7}$-NP-induced Tet+ and KLH-induced TFH cells as compared to Tconv cells. Notably, most of these differentially methylated enhancers (66%) were shared by Tet+ and TFH cells. As with most other readouts examined herein, there was no statistically significant association between the presence of differentially hypo- or hypermethylated active enhancers and overall gene expression differences (*Figure 7B*; p=1) (*Figure 7—source data 1*). This observation is consistent with the idea that the methylation status of distal GREs for genes specifically upregulated in TR1 cells is almost invariably imprinted at the TFH stage, thus indicating that TFHs are epigenetically poised to acquire a TR1 transcriptional profile. The *Il10* locus, for example, whose expression is significantly upregulated in BDC2.5mi/I-A$^{g7}$-NP-induced Tet+ cells as compared to KLH-induced TFH cells, harbors eight and six differentially hypomethylated active enhancers in Tet+ and TFH samples, respectively, as compared to Tconv cells (*Figure 6C*). Thus, most of the upregulated genes at the TR1 stage appear to have already demethylated their distal GREs at the TFH stage.

## Active enhancers at genes specifically upregulated at the TR1 cell stage are enriched for binding sites for the TFH/TR1 TFs TOX-2, IRF4, and c-MAF

We next sought to investigate if the putative active enhancers identified in BDC2.5mi/I-A$^{g7}$-NP-induced Tet+ and KLH-induced TFH cells contained binding sites for the TFH TFs TOX-2, IRF4, and c-MAF. The genes that were upregulated at the TR1 cell stage and shared an open chromatin status with TFH cells, based on the scMultiome dataset, were associated with 285 active enhancers. We then mapped ChIPseq peaks for TOX-2 from TFH cells, and IRF4 and c-MAF from Th17 cells (*Ciofani et al., 2012*; *Li et al., 2012*; *Xu et al., 2019*) onto these active enhancers. Remarkably, 34.4%, 66.3%, and 70.1% of these enhancers had TOX-2, IRF4, and c-MAF binding sites, respectively, and ~61% of these TF-binding active enhancers are already present in TFH cells (vs. <2% in Tconv cells; p<3.03e-8) (*Figure 7C* and *Figure 7—source data 2*).

Thus, many of the active enhancers linked to genes specifically upregulated at the TR1 cell stage, which, in turn, are highly hypomethylated and accessible at the TFH cell stage, are enriched for binding sites for all the three TFH TFs studied herein: TOX-2, IRF4, and c-MAF. Therefore, these sites are likely already occupied by these TFs at the TFH stage, especially considering their described role in TFH development and maintenance. For example, many hypomethylated active enhancers linked to *Il10* in BDC2.5mi/I-A$^{g7}$-NP-induced Tet+ cells and KLH-induced TFH cells have binding sites for TOX-2 (as well as IRF4 and c-MAF) in TFH and Th17 cells (*Figure 6C*).

## Discussion

Having established that TFH cells can transdifferentiate into TR1-like and terminally differentiated TR1 cells in vivo in response to certain stimuli (*Solé et al., 2023b*; *Solé et al., 2023a*), we sought to explore the epigenetic events underpinning this process. Our comprehensive transcriptional and epigenetic studies at the bulk and/or single-cell levels indicate that conventional antigen experienced TFH cells are epigenetically poised to become TR1 cells. One of our main findings is that the TFH-to-TR1 differentiation process is associated with massive closure of OCRs, and that the vast majority of the OCRs found in TR1-like and TR1 cells are also found in KLH-DNP-induced TFH cells. Furthermore, most of the genes harbored in these shared OCRs, such as *Cxcr5, Tox2, Il21, Il10,* and *Ctla4*, to just name a few, contain nearly identical patterns of histone deposition marks, including H3K4me3, H3K27me3, and H3K27Ac, equally similar DNA methylation patterns in gene bodies, proximal promoters, and gene-distal regulatory elements and a similar distribution of active enhancers across the genome, even when not expressed in either cell type. Altogether, these data indicate that the TR1-poised epigenome of TFH cells is a key enabler of this transdifferentiation process, and that transdifferentiation of TFH cells into TR1 cells is likely driven by changes in the expression of TFH-stabilizing and TR1-promoting TFs, possibly in response to sustained ligation of TCRs.

Our earlier scRNAseq and mass cytometry studies of cognate (antigen-specific, tetramer+) pMHCII-NP-induced CD4+ T cells demonstrated the presence of a significant Tet+ TFH-like cluster that

separated away from its TR1 counterpart (*Solé et al., 2023b*; *Solé et al., 2023a*). When compared to each other, these two major clusters of tetramer+ cells were remarkably similar, but the TR1 sub-pool had significantly downregulated key TFH-specific genes, including *S1pr2*, *Cxcr4*, *Cxcr5*, *Pdcd1*, *Il4*, *Ascl2*, *Bcl6*, *Cba2t3*, *Cebpa*, *Id3*, *Nfia*, *Pou2af1*, *Tox2*, while upregulating TR1-associated genes, such as *Ccr5*, *Havcr2*, *Il10*, *Ahr*, *Myc*, and *Prdm1*. Importantly, these two clusters were developmentally related because they harbored identical clonotypes (i.e. identical TCRαβ sequences at different transcriptional states). Additional studies indicated that the TFH-to-TR1 conversion proceeds through a transitional TR1-like subset, whereby the progenitors undergo progressive downregulation of TFH-associated transcripts (i.e. *Bcl6*, *Cxcr5*, among others) and progressive upregulation of TR1-associated transcripts (i.e. *il10*, *Ccr5*, and *Prdm1*, among others). The suspected TFH origin of pMHCII-NP-induced TR1 cells was further supported by two additional lines of evidence. First, treatment of NOD.*Scid* mice engrafted with total CXCR5$^{high}$PD-1$^{high}$ CD4+ T-cells (containing pMHCII-NP-expanded TFH-like cells but devoid of terminally differentiated Tet+ TR1 cells) with pMHCII-NPs led to the formation of cognate TR1-like and terminally differentiated TR1 cell pools in the hosts. Second, the two-dimensional scMultiome profile of Tet+ TFH-like cells arising in response to BDC2.5mi/IA$^{g7}$-NP therapy was essentially identical to that corresponding to an effector TFH-like sub-pool of TFH-like cells (CD4$^+$CD44$^{hi}$CXCR5$^{hi}$PD1$^{hi}$) induced by immunization with the KLH-DNP conjugate (*Bcl6$^{hi}$Tox-2$^{hi}$Il21$^+$Pdcd1$^+$*; referred to as TFH.1).

Abrogation of *Prdm1* (encoding BLIMP-1) expression enabled the conversion of TFH.1 cells into TR1-like progeny, but completely blunted the TR1-like–>TR1 conversion, indicating that this process requires the expression of BLIMP-1, a transcriptional repressor that antagonizes BCL-6 expression and function in both B- and T-cells, including TFH cells (*Crotty et al., 2010*). Since expression of BLIMP-1 in T-cells is restricted to activated T-cells and is induced by TCR ligation, and since pMHCII-NP therapy triggers the formation and expansion of cognate, antigen-specific TR1-like cell pools via sustained TCR signaling (*Singha et al., 2017*), we suspect that BLIMP-1 expression in TFH cells is induced by repetitive encounters of cognate TFH cells with these compounds. Progressive downregulation of *Bcl6* and upregulation of *Prdm1* at the TR1-like cell stage, immediately preceding TR1 development, might be facilitated by the loss of *Lef1* and significant downregulation of *Tcf7* (encoding TCF-1) expression (positive regulators of *Bcl6* expression and negative regulators of *Prdm1* expression) in pMHCII-NP-challenged vs. vaccine-induced TFH cells. In fact, the Tet+ TR1-like and terminally differentiated TR1 cells (BLIMP-1-independent or -dependent, respectively) differ in the expression levels of a significant number of genes whose expression has been previously associated with this TF, including *Il10*, *Ctla4*, *Lag3*, *Icos*, *Havcr2*, *Tnfrsf4*, and *Tnfrsf18*, among others (*Chihara et al., 2018*).

Here, we have shown that pMHCII-NP-driven transdifferentiation of cognate TFH cells into TR1 progeny is driven by both massive closure of OCRs and major changes in the transcriptional factor make-up of the cells, rather than by TCR signaling-induced changes in the epigenetic status of the genes contained within the shared OCRs. There are some noteworthy differences between the evolution of the epigenetic landscape in effector, memory, and exhausted T-cells as compared to what we see in the TFH-TR1 pathway. In T-cells, acquisition of the epigenetic programs of effector, memory, and exhaustion states are stepwise processes. In exhausted T-cells, for example, ATACseq and H3K27AC ChIPseq analyses of different exhausted states are associated with distinct epigenetic states, including differential chromatin accessibility and active enhancer landscapes, yet shared TF profiles (*Belk et al., 2022*). In contrast, transdifferentiation of TFH cells into TR1 cells appears to be driven by changes in the expression of TFH-stabilizing and TR1-promoting TFs, in the face of remarkable epigenetic similarity, including a shared methylome, consistent with the idea that TFH cells are poised to rapidly become TR1 when appropriately activated. Interestingly, the TF-coding genes that are significantly downregulated during the TFH-to-TR1 conversion, unlike those that are upregulated, experience a significant reduction in chromatin accessibility. In contrast, as expected based on the epigenetic similarity of TFH vs. TR1 cells, upregulation of TF-coding genes important for TR1 cell genesis is largely dissociated from the different epigenetic marks studied here, including changes in chromatin accessibility. This suggests that changes in TF expression, particularly the loss of TFH-associated TFs, are driven, in part, via chromatin remodeling of the coding loci.

The combination of TFs that become available (i.e. *Prdm1*) or unavailable (i.e. *Bcl6*) during the TFH-TR1 conversion would be responsible for enabling/driving the formation of the active enhancer-promoter contacts required for gene expression and establishing TR1 transcriptional identity. This

would explain why TR1-associated genes that are already epigenetically marked for expression in TFH cells (i.e. *Il10*) remain silent at the TFH stage. This is not a unique feature of the TFH-to-TR1 transdifferentiation process. For example, although the *Il10* locus is closed and transcriptionally silent in naive CD4+ T-cells, it is exposed (*Miraldi et al., 2019*) and incorporates permissive H3K4me3 marks but not repressive H3K27me3 marks in differentiated T helper subsets (*Ciofani et al., 2012*; *Wei et al., 2010*), promoting a transcriptional competent yet silent state. Since TFs regulating Th1, Th2, and Th17 subsets, including T-bet, GATA-3, or RORγt, do no inhibit IL-10 expression, but rather enhance it, the above observations imply that different effector cell programs can co-exist with much broader gene expression-competent states. Our data indicate that this is also true for TFH cells. In fact, the expression of *Il10* in Th cell subsets can only occur until the TFs required for gene expression become available. For example, BATF (an AP1 family member), IRF4, NFAT, c-MAF, AHR, and BLIMP-1 are known to collectively promote *Il10* expression by binding to the promoter and/or cis-regulatory elements of *Il10* (*Pot et al., 2011*). Likewise, although Th lineage-specific cytokines present active histone marks in the corresponding lineage and repressive marks in the others, TF-coding genes are not always so strictly marked and display expression-permissive epigenetic patterns. For example, *Tbx21*, encoding T-bet, harbors activating H3K4me3 marks in the promoter of Th1 cells, but bivalent modifications in other Th subsets. This is also true for *Gata3* and *Rorc* or *Bcl6* in Th2 vs. non-Th2, Th17 vs. non-Th17, and TFH vs. non-TFH cells, respectively (*Lu et al., 2011*; *Wei et al., 2009*). Thus, whereas *Bcl6* is also marked with expression-promoting H3K4me3 marks in other Th cell subsets, loci encoding Th subset-specifying TFs (*Tbx21*, *Gata3,* and *Rorc*) also appear to be poised for expression in TFH cells (*Lu et al., 2011*). Because BCL-6 can regulate the expression other TFs, it is likely the collective TF make-up of the cell that determines its phenotype at a given point in time. It is thus reasonable to suspect that the epigenomes of polarized T-cell subsets, including TFH cells, are programmed to enable their conversion into alternative transcriptional states when required (i.e. in response to excessive antigenic stimulation).

The TR1-poised state of TFH cells is also reflected on the global active enhancer landscapes of both cell types. For example, we have shown that most of the genes that remain open as BDC2.5mi/I-A$^{g7}$-NP-induced TFH.1 cells transition into TR1-like/TR1 cells also share active enhancers in both subsets. In addition, the demethylated status of most active enhancers associated with genes specifically upregulated in TR1 cells, such as *Il10*, were already so (hence imprinted) at the TFH stage. Furthermore, we find that many of the active enhancers linked to genes specifically upregulated at the TR1 cell stage, which, in turn, are highly hypomethylated and accessible at the TFH cell stage, are enriched for binding sites for at least three TFH-associated TFs, including TOX2, IRF4, and c-MAF. This suggests that these sites are likely already occupied by these TFs at the TFH stage. The remarkable similarity of the DNA methylome of the TFH and TR1 subsets further suggests that both TFH and TR1 cells share a stable epigenetic program.

The factors responsible for chromatin closure during the TFH-to-TR1 conversion remain to be determined, but changes in the expression of *Tox2* (and the related TF *Tox*) may contribute to this event. TOX-2 expression in CD4+ T-cells is upregulated by BCL-6, and TOX-2 binds to loci associated with TFH generation, including *Bcl6*, *Cxcr5*, *Pdcd1* (also upregulated in TR1-like cells) along with BATF and IRF4 or STAT-3, increasing their chromatin accessibility and promoting their expression (*Xu et al., 2019*). Importantly, whereas KLH-DNP- and pMHCII-NP-induced TFH cells express high levels of *Tox2*, their TR1-like and TR1 counterparts express substantially lower levels; suboptimal occupation of TOX-2 binding sites by TOX-2 in pMHCII-NP-challenged TFH cells may result in the closure of these sites, cessation of the expression of the corresponding genes and progressive differentiation into TR1 cells. Downregulation of TCF-1 (encoded by *Tcf7*) and LEF-1 expression as TFH cells become TR1 cells may be additional contributing factors to loss of chromatin accessibility (*Gounari and Khazaie, 2022*). We note that loss of TCF-1 in T-cell subsets is usually paralleled by loss of LEF-1, which belongs to the same TF family and recognizes a similar motif (*Gounari and Khazaie, 2022*). In developing thymocytes, upregulation of TCF-1 expression is associated with an increase in chromatin accessibility, suggesting that it may act as a pioneer TF. Furthermore, loss of TCF-1 in DP thymocytes or CD8+ T-cells, T-exhausted stem cells or activated CD4+ or CD8+ T-cells is associated with loss of chromatin accessibility at sites that had bound TCF-1 (*Gounari and Khazaie, 2022*). In addition, TCF-1, along with CTCF, promotes deposition of H3K27ac onto insulated enhancers and the recruitment of cohesin-loading factor NIPBL at active enhancers in developing thymocytes (*Wang et al., 2022*). In

another recent study, TCF-1 was identified as a 'placeholder' TF, responsible for maintaining chromatin accessibility in naïve T-cells, and allowing activation-induced TFs to displace it (*Zhong et al., 2022*). Hence, downregulation of TCF-1 and LEF-1 may contribute to the loss of the TFH-specific gene expression program as these cells transdifferentiate into TR1, close previously open chromatin sites, and acquire new TFs orchestrating the TR1-specific gene expression program.

The current study provides a foundational understanding of how the epigenetic landscape of TFH cells evolves as they transdifferentiate into TR1 progeny in response to chronic ligation of cognate TCRs using pMHCII-NPs. Our current studies focus on functional validation of these observations, by carrying out extensive perturbation studies of the TFH-TR1 transdifferentiation pathway in conditional TF gene knockout mice. In these ongoing studies, genes coding for a series of TFs expressed along the TFH-TR1 pathway are selectively knocked out in T cells, to ascertain (1) the specific roles of key TFs in the various cell conversion events and transcriptional changes that take place along the TFH-TR1 cell axis; (2) the roles that such TFs play in the chromatin re-modeling events that underpin the TFH-TR1 transdifferentiation process; and (3) the effects of TF gene deletion on phenotypic and functional readouts of TFH and Treg function. Although the TFH-TR1 transdifferentiation was discovered in mice treated with pMHCII-NPs, we now have evidence that this is a naturally occurring pathway that also develops in other contexts (i.e. in mice that have not been treated with pMHCII-NPs). Importantly, the discovery of this transdifferentiation process affords a unique opportunity to further understand the transcriptional and epigenetic mechanisms underpinning T-cell plasticity; the findings reported here can help guide/inform not only upcoming translational studies of pMHCII-NP therapy in humans, but also other research in this area.

Although the snapshot provided by our single-cell studies reported herein documents the simultaneous presence of the different subsets composing the TFH-TR1 cell pathway upon the termination of treatment, the transdifferentiation process itself is extremely fast, such that proliferated TFH cells already transdifferentiate into TR1 cells after a single pMHCII-NP dose (*Solé et al., 2023b*). This makes it extremely challenging to pursue dynamic experiments. Notwithstanding this caveat, ongoing studies of cognate T-cells post treatment withdrawal, coupled to single-cell studies of the TFH-TR1 pathway in TF gene knockout mice exhibiting perturbed transdifferentiation processes, are likely to shed light into the progression and stability of the epigenetic changes reported herein.

We have recently shown that αGalCer/CD1d-NPs can trigger the differentiation of liver-resident invariant NKT cells (LiNKT) into a TR1-like immunoregulatory, IL-10+IL-21-producing Zbtb16[high]-Maf[high]Tbx21+Gata3+Rorc− subset (LiNKTR1) that can suppress local inflammatory phenomena (*Umeshappa et al., 2022*). Interestingly, epigenetic studies of liver iNKT cells both before and after in vivo delivery of aGalCer/CD1d-coated NPs have shown that unlike the case for pMHCII-NP-induced TR1 transdifferentiation, aGalCer/CD1d-NP-induced LiNKTR1 transdifferentiation involves the acquisition of a novel epigenetic state. Specifically, whereas for most genes, gene upregulation during the LinKT-to-LiNKTR1 cell transition is largely associated with treatment-induced hypomethylation, the most upregulated genes (i.e. *Il10* and *Il21*, among others) are also those that accumulate additional epigenetic modifications favoring gene expression, such as acquisition of new OCRs and H3K27ac and H3K4me3 marks. Thus, whereas TFH cells largely require chromatin closure and changes in TF expression to become TR1 cells, LiNKT cells do not undergo massive changes in chromatin exposure and involve extensive gene demethylation to do so. Although the mechanisms underlying these differences remain unclear, they indicate that the processes that regulate responses of different T-cell types to similar cues are context-dependent and dynamic.

# Materials and methods

## Key resources table

| Reagent type (species) or resource | Designation | Source or reference | Identifiers | Additional information |
|---|---|---|---|---|
| Gene (*Mus musculus*) | Mouse reference genome | UCSC; http://hgdownload.soe.ucsc.edu/goldenPath/mm10/bigZips/ | Mouse reference genome NCBI build 38, GRCm38/mm10 | |

*Continued on next page*

*Continued*

| Reagent type (species) or resource | Designation | Source or reference | Identifiers | Additional information |
|---|---|---|---|---|
| Strain, strain background (*Mus musculus*) | Mouse: NOD/ShiLtJ | The Jackson Laboratory | Cat#001976 | |
| Cell line (Cricetulus griseus) | Chinese hamster: CHO-S cells | ThermoFisher | Cat#R80007 | |
| Antibody | Monoclonal Anti-mouse CD4-FITC | BD Biosciences | Cat#553047 | 5 ug/ml |
| Antibody | Monoclonal Anti-mouse CD4-PB | BioLegend | Cat#100428 | 5 ug/ml |
| Antibody | Monoclonal Anti-mouse CD45R/ B220-PerCP | BD Biosciences | Cat#553093 | 2 ug/ml |
| Antibody | Monoclonal Anti-mouse CD16/32 | BD Biosciences | Cat#553141 | 1/100 |
| Antibody | Monoclonal Anti-mouse CD16/32 | BioLegend | Cat#101320 | 1/100 |
| Antibody | Monoclonal Anti-mouse CXCR5-biotin | BD Biosciences | Cat#551960 | 1/100 |
| Antibody | Monoclonal Anti-mouse CD44-FITC | BD Biosciences | Cat#553133 | 1/100 |
| Antibody | Monoclonal Anti-mouse CD279 (PD1) | BD Biosciences | Cat#562584 | 1/100 |
| Antibody | Polyclonal Anti-mouse Histone 3 (acetyl K27) | Abcam | Cat#ab4729 | 1/200 |
| Antibody | Polyclonal Anti-mouse Histone 3 (trimethyl K4) | Sigma-Aldrich | Cat#04–745 | 1/1200 |
| Antibody | Polyclonal Anti-mouse Histone 3 (trimethyl K27) | CellSignaling | Cat#9733 S | 1/1200 |
| Sequence-based reagent | Ad1_noMx | Conda | | |
| Sequence-based reagent | Ad2.1 | Conda | | Ad2.1: TAAGGCGA |
| Sequence-based reagent | Ad2.2 | Conda | | Ad2.2: CGTACTAG |
| Sequence-based reagent | Ad2.3 | Conda | | Ad2.3: AGGCAGAA |
| Sequence-based reagent | Ad2.4 | Conda | | Ad2.4: TCCTGAGC |
| Sequence-based reagent | Ad2.5 | Conda | | Ad2.5: GGACTCCT |
| Sequence-based reagent | Ad2.6 | Conda | | Ad2.6: TAGGCATG |
| Recombinant protein | BDC2.5mi/IAg7 tetramer | *Clemente-Casares et al., 2016* | Not applicable | |
| Recombinant protein, fluorochrome labelled | Streptavidin-PE | Life Technologies | Cat#SNN1007 | 1:4 molar ratio with pMHCII monomers |
| Commercial assay or kit | Anti-mCD4 microbeads | Milteny Biotec | Cat#130-049-201 | |
| Commercial assay or kit | Nextera XT | Illumina | Cat#FC-131–1024 | |
| Commercial assay or kit | RNeasy Plus Mini Kit | Qiagen | Cat#74134 | |

*Continued on next page*

*Continued*

| Reagent type (species) or resource | Designation | Source or reference | Identifiers | Additional information |
|---|---|---|---|---|
| Commercial assay or kit | TruSeq Stranded mRNA Sample Prep Kit v2 | Illumina | Cat#RS-122-2101/2 | |
| Commercial assay or kit | Reverse transcriptase SuperScript II | Invitrogen | Cat#18064–014 | |
| Commercial assay or kit | Agencourt AMPure XP Beads | Beckman Coulter | Cat#A63881 | |
| Commercial assay or kit | KAPA Library Quantification Kit | KapaBiosystems | Cat#KK4835 | |
| Commercial assay or kit | BirA500 biotinylation kit | Avidity | Cat#BirA500 | |
| Commercial assay or kit | EZ DNA Methylation-Gold kit | ZYMO | Cat#D5005 | |
| Commercial assay or kit | MinElute PCR purification kit | Qiagen | Cat#28004 | |
| Commercial assay or kit | MinElute Reaction Cleanup kit | Qiagen | Cat#28204 | |
| Commercial assay or kit | NEBNext High Fidelity PCR kit | New England BioLabs | Cat#M0541S | |
| Commercial assay or kit | NEBNext Ultra DNA Library Prep kit | Illumina | Cat#E7645S | |
| Commercial assay or kit | Nextera DNA Library Preparation kit | Illumina | Cat#FC-121–1030 | |
| Commercial assay or kit | Pierce Comassie (Bradford) Kit | Thermo Fisher Scientific | Cat#23200 | |
| Commercial assay or kit | Cat#23200 | Thermo Fisher Scientific | Pierce Comassie (Bradford) Kit | |
| Commercial assay or kit | Pierce Monomeric Avidin Kit | Thermo Fisher Scientific | Cat#20227 | |
| Commercial assay or kit | Tagment DNA TDE1 Enzyme and Buffer kit | Illumina | Cat#20034197 | |
| Commercial assay or kit | PCR cleanup | Qiagen | Cat#28104 | |
| Commercial assay or kit | Dynabeads Protein A | Invitrogen | Cat#10001D | |
| Commercial assay or kit | MACS separation LS columns | Miltenyi Biotec | Cat#130-042-401 | |
| Commercial assay or kit | Chromium Next GEM single cell 3' reagent v3.1 | 10 x Genomics | Cat#PN-1000128 | |
| Commercial assay or kit | Amicon Ultra-15 100 kDa cut-off | Millipore | Cat#UFC910024 | |
| Commercial assay or kit | DNA LoBind 1.5 ml tubes | Eppendorf | Cat#0030108051 | |
| Commercial assay or kit | PD-10 Desalting Columns | GE Healthcare | Cat#52-1308-00 BB | |
| Commercial assay or kit | Pierce BCA Assay Kit | Thermo Fisher Scientific | Cat#23225 | |
| Commercial assay or kit | Cat#23225 | Thermo Fisher Scientific | Pierce BCA Assay Kit | |

*Continued on next page*

*Continued*

| Reagent type (species) or resource | Designation | Source or reference | Identifiers | Additional information |
|---|---|---|---|---|
| Commercial assay or kit | Protein LoBind 1.5 ml tubes | Eppendorf | Cat#0030108442 | |
| Commercial assay or kit | Ultra-Fine 30 G insulin syringes | BD | Cat#320927 | |
| Chemical compound, drug | L-Glutamine Solution (200 mM) | Cultek | Cat#H3BE17-605E | |
| Chemical compound, drug | Penicillin/Streptomycin | Sigma-Aldrich | Cat#P4333 | |
| Chemical compound, drug | Gentamycin | Lonza | Cat#91 L0012-010 | |
| Chemical compound, drug | Keyhole Limpet Hemocyanin (KLH) | Sigma-Aldrich | Cat#H7017 | |
| Chemical compound, drug | DNP-Keyhole Limpet Hemocyanin Conjugate, (DNP-KLH) | Sigma-Aldrich | Cat#324121 | |
| Chemical compound, drug | Freund's Adjuvant, Complete | Sigma-Aldrich | Cat#F5881-10ML | |
| Chemical compound, drug | Freund's Adjuvant, Incomplete | Sigma-Aldrich | Cat#F5506-10ML | |
| Chemical compound, drug | Avidin | Thermo Scientific | Cat#21121 | |
| Chemical compound, drug | Maleimide-PEG 2 kDa | Jenkem Tech | Cat#MAL-PEG2000-MAL | |
| Chemical compound, drug | Cell Boost 7 a | HyClone | Cat#SH31026.07 | |
| Chemical compound, drug | Cell Boost 7b | HyClone | Cat#SH31027.01 | |
| Chemical compound, drug | Dulbecco's Phosphate Buffered Saline (DPBS) | Sigma-Aldrich | Cat#D8573-1L | |
| Chemical compound, drug | Paraformaldehyde | Electron Microscopy Sciences | Cat#15710 | |
| Chemical compound, drug | Protease Inhibitor Cocktail | Roche | Cat#4693132001 | |
| Chemical compound, drug | Proteinase K | Roche | Cat#03 115 879 001 | |
| Chemical compound, drug | DMEM | Sigma-Aldrich | Cat#D6429−6X1 L | |
| Chemical compound, drug | RNAse A | Qiagen | Cat# 19101 | |
| Chemical compound, drug | Fetal Bovine Serum (FBS) | Sigma-Aldrich | Cat#F7524 | |
| Software, algorithm | BiocManager (v1.30.16) | https://cran.r-project.org/package=BiocManager%0A%0A | | |
| Software, algorithm | biomaRt (v2.48.3) | *Durinck et al., 2009* | | |
| Software, algorithm | Bowtie2 (v2.4.2) | *Langmead and Salzberg, 2012* | | |
| Software, algorithm | BSMAP (v3.0) | *Xi and Li, 2009* | | |
| Software, algorithm | BWA (v0.0.) | *Li and Durbin, 2010* | | |

*Continued on next page*

*Continued*

| Reagent type (species) or resource | Designation | Source or reference | Identifiers | Additional information |
|---|---|---|---|---|
| Software, algorithm | Cellranger (v6.0) | https://support.10xgenomics.com/single-cell-gene-expression/software/pipelines/latest/what-is-cell-ranger | | |
| Software, algorithm | Cellranger-arc (v2.0) | https://support.10xgenomics.com/single-cell-multiome-atac-gex/software/pipelines/latest/what-is-cell-ranger-arc | | |
| Software, algorithm | Cellranger-atac (v2.1) | https://support.10xgenomics.com/single-cell-atac/software/pipelines/latest/what-is-cell-ranger-atac | | |
| Software, algorithm | ChipSeeker (v1.28.3) | *Yu et al., 2015* | | |
| Software, algorithm | clusterProfiler (v4.0.5) | *Wu et al., 2021* | | |
| Software, algorithm | Deeptools (v3.5.0) | *Ramírez et al., 2014* | | |
| Software, algorithm | Deseq2 (v1.32.0) | *Love et al., 2014* | | |
| Software, algorithm | DiffBind (v3.2.7) | *Ross-Innes et al., 2012* | | |
| Software, algorithm | FastQC | http://www.bioinformatics.babraham.ac.uk/projects/fastqc/ | | |
| Software, algorithm | FlowJo v9 | Becton, Dickinson and Company; https://www.flowjo.com | | |
| Software, algorithm | Gene Ontology | http://geneontology.org | | |
| Software, algorithm | Genomic Ranges (v1.44.0) | *Lawrence et al., 2013* | | |
| Software, algorithm | MACS2 (v2.2.7.1) | *Zhang et al., 2008* | | |
| Software, algorithm | Monocle3 (v1.0.1) | *Cao et al., 2019* | | |
| Software, algorithm | org.Mm.eg.db (v3.13.0) | https://bioconductor.org/packages/release/data/annotation/html/org.Mm.eg.db.html | | |
| Software, algorithm | Partek Flow software | https://www.partek.com/partek-flow/ | | |
| Software, algorithm | Picard (v2.25.0) | https://broadinstitute.github.io/picard/ | | |
| Software, algorithm | R (v4.1.0) | https://www.eea.europa.eu/data-and-maps/indicators/oxygen-consuming-substances-in-rivers/r-development-core-team-2006 | | |
| Software, algorithm | Rstudio (v1.4.1103) | https://www.rstudio.com/ | | |
| Software, algorithm | R- trackViewer Bioconductor package | https://github.com/jianhong/trackViewer (*jianhong, 2024*) | | |
| Software, algorithm | Samtools | http://samtools.sourceforge.net | | |
| Software, algorithm | Seurat (v4.0.3) | *Hao et al., 2021* | | |
| Software, algorithm | Signac (v1.3.0) | *Stuart et al., 2021* | | |
| Software, algorithm | STAR (v2.7.10a) | *Dobin et al., 2013* | | |
| Software, algorithm | Tidyverse (v1.3.1) | https://www.tidyverse.org | | |

*Continued on next page*

*Continued*

| Reagent type (species) or resource | Designation | Source or reference | Identifiers | Additional information |
|---|---|---|---|---|
| Software, algorithm | Trackviewer (v1.31.1) | *Ou and Zhu, 2019* | | |
| Software, algorithm | Trimmomatic (v.03) | *Bolger et al., 2014* | | |
| Other | ÄKTA Pure 25 FPLC | GE Healthcare | | FPLC equipment used to purify proteins |
| Other | BD FACSAria II | BD Biosciences | | Flow cytometer |
| Other | BD FACSAria SORP | BD Biosciences | | Flow cytometer |
| Other | Bioruptor | Diagenode | | Sonicator |
| Other | Covaris S220 | Covaris | | Sonicator |
| Other | Zetasizer dynamic light scatter (DLS) equipment | Malvern | | Equipment used to measure hydrodynamic diameter and monodispersion of nanoparticles |
| Other | HiSeq2500 | Illumina | | DNA sequencing apparatus |
| Other | NovaSeq 6000 | Illumina | | DNA sequencing apparatus |
| Other | Qubit fluorometer | Invitrogen | | DNA/RNA/protein quantification apparatus |
| Other | TapeStation 4200 | Agilent | | Automated electrophoresis apparatus for DNA/RNA sample quality control |

## Mice

NOD/ShiLtJ mice (strain #: 001976) were from the Jackson Lab (Bar Harbor, ME, USA). The experiments described herein were approved by the Cumming School of Medicine's Animal Care Committee at the University of Calgary (protocol AC24-0053) and by the Animal Care Committee at Universitat de Barcelona (protocol 130/19).

## pMHCII production

Recombinant pMHC class II were produced in CHO-S cells transduced with lentiviruses encoding peptide-MHCII beta and MHCII alpha chains and IRES-CFP and IRES-EGFP cassettes, respectively, as described (*Serra et al., 2019*). Briefly, transduced CHO cells were grown in 2 L baffled flasks (Nalgene, Thermo Fisher Scientific, Waltham, MA, USA) at 125 rpm, 5% $CO_2$ and 37°C. Basal medium was Power-CHO-2 (Lonza, Basel, Switzerland) supplemented with 8 mM Glutamine (Cultek, Madrid, Spain) and Gentamicine Sulfate (0.25 mg/mL) (Lonza). The cultures were started in a volume of 400 mL of basal medium at a cell density of 350,000–400,000 cells/mL and were supplemented with Cell Boost 7a (Hyclone) at 3% vol/vol and Cell Boost 7b (Hyclone, GE Healthcare, Chicago, IL, USA) at 0.3% vol/vol on days 0, 3, 4, 5, 6, 8, 9, and 10. Temperature shift to 34°C was done when cell densities reached $5–7×10^6$ cells/mL. Additional Glutamine was added on day 7,–2 mM. Glucose was added to 4.5 g/L when levels dropped below 3.5 g/L. Cells were harvested on day 14 or when viability fell below 60%. The secreted proteins were purified by sequential affinity chromatography on nickel and strep-tactin columns and used for NP coating or biotinylated in vitro to produce pMHCII tetramers.

## pMHCII tetramers

Phycoerythrin (PE)- or APC-conjugated tetramers were prepared using biotinylated pMHCII monomers and used to stain peripheral T-cells. Briefly, pMHCII monomers were subjected to biotinylation using biotin ligase (Avidity, Aurora, CO, USA) following the supplier's protocols, and biotinylated monomers purified by ion exchange chromatography using an AKTA FPLC system (GE Healthcare, Chicago, IL, USA). The final product was verified by denaturing SDS-PAGE. Tetramers were generated by adding PE-conjugated streptavidin (Life Technologies, Carlsbad, CA, USA) at a 4:1 molar ratio.

## Flow cytometry for pMHCII-NP-induced Tet+ cells and KLH-DNP-induced TFH cells

To stain mononuclear cell suspensions from mice, splenic CD4+ T-cells were incubated with avidin for 15 min at room temperature and stained with tetramer (5 µg/mL) in FACS buffer (0.05% sodium azide and 1% FBS in PBS) for 30 min at 4°C, washed, and incubated with FITC-conjugated anti-CD4 (RM4-5 or GK1.5 from BD Biosciences, San Diego, CA, USA; 5 µg/mL) and PerCP-conjugated anti-B220 (RA3-6B2 from BD Biosciences; 2 µg/mL; as a 'dump' channel) for 30 min at 4°C, in the presence of an anti-CD16/CD32 mAb (2.4G2; BD Biosciences, or BioLegend, San Diego, CA, USA) to block Fc receptors. Cells were washed, fixed in 1% paraformaldehyde (PFA) in PBS, and analyzed with FACSaria, or BD LSRII flow cytometers. Analysis was done using FlowJo software (FlowJo, BD Biosciences, San Diego, CA, USA).

TFH cells (PD-1$^{hi}$CXCR5$^{hi}$) were generated by immunizing NOD mice intraperitoneally with KLH or KLH-DNP (Sigma-Aldrich, St. Louis, MO, USA) three times (100 µg/dose, CFA+IFA+IFA) once a week for 3 consecutive weeks. Splenic T-cells were stained with anti-CD4-Pacific Blue (GK1.5, BD Biosciences), anti-CD45R-PerCP (BD Biosciences), anti-CD44-FITC (IM7 from BD Biosciences), anti-CXCR5-biotin (2G8 from BD Biosciences), and anti-CD279-BV421 (PD-1; J43 from BD Biosciences) mAbs for 30 min at 4°C and with streptavidin-APC for 20 min at 4°C. TFH cells were identified within the CD4+CD45R– CD44$^{hi}$ gate as cells expressing high levels of CXCR5 and CD279 (PD-1).

## NP synthesis

Maleimide-functionalized, pegylated iron oxide NPs (PFM series) were produced in a single-step thermal decomposition in the absence of surfactants as described recently (*Singha et al., 2017*). Briefly, 3 g Maleimide-PEG (2 kDa MW, Jenkem Tech USA) were melted in a 50 mL round-bottom flask at 100°C and then mixed with 7 mL of benzyl ether and 2 mmol Fe(acac)$_3$. The reaction was stirred for 1 hr and heated to 260°C with reflux for 2 hr. The mixture was cooled to room temperature and mixed with 30 mL water. Insoluble materials were removed by centrifugation at 2000×*g* for 30 min. The NPs were purified using magnetic (MACS) columns (Miltenyi Biotec, Auburn, CA, USA) and stored in water at room temperature or 4°C. The concentration of iron was determined spectrophotometrically at 410 nm in 2 N hydrochloric acid (HCl).

## pMHCII conjugation to NPs

pMHCII conjugation to maleimide-functionalized NPs (PFM) was done via the free C-terminal Cys engineered into the MHCII chain/knob. Briefly, pMHCs were mixed with NPs in 40 mM phosphate buffer, pH 6.0, containing 2 mM ethylenediaminetetraacetic acid, 150 mM NaCl, and incubated overnight at room temperature. pMHCII-conjugated NPs were purified by magnetic separation and concentrated by ultrafiltration through Amicon Ultra-15 (100 kDa cut-off) (Merck KGaA, Darmstadt, Germany) and stored in PBS.

## NP characterization

The size and dispersity of unconjugated and pMHCII-conjugated NPs were assessed via transmission electron microscopy (TEM, Hitachi H7650, Hitachi, Chiyoda, Tokyo, Japan) and dynamic light scattering (DLS, Zetasizer, Malvern Panalytical, Spectris, Egham, UK). Pegylated and pMHC-NPs were analyzed via 0.8% agarose gel electrophoresis, native and denaturing 10% SDS-PAGE. To quantify pMHCII valency, we measured the pMHCII concentration of the pMHCII-NP preps using the Bradford assay (Thermo Scientific).

## pMHCII-NP therapy of NOD mice

Cohorts of 10-week-old female NOD mice were injected i.v. with BDC2.5mi/IA$^{g7}$-coated NPs in PBS twice a week for 5 weeks. Treatment-induced formation and expansion of cognate tetramer+ CD4+ T-cells were assessed by flow cytometry.

## CD4+ T-cell samples used for next-generation sequencing

Unless stated otherwise, next-generation sequencing data was obtained from non-restimulated BDC2.5/IA$^{g7}$-NP-induced CD4+ BDC2.5/IA$^{g7}$-tetramer$^+$ (CD4$^+$/B220$^-$/tet$^+$) T-cells, Tconv (CD4$^+$/B220$^-$/tet$^-$) cells, also obtained from BDC2.5/IA$^{g7}$-NP-treated mice; KLH-DNP-immunized TFH (CD4$^+$/B220$^-$/

CD44[+]/PD1[+]/CXCR5[high]), and TH0 cells obtained from KLH-DNP-immunized mice (CD4[+]/B220[−]/CD44[−]/PD1[−]/CXCR5[−]).

## Bulk RNAseq

Cells were sorted in lysis buffer or PBS (5e4 cells) to perform RNA extractions for RNAseq. For bulk RNAseq, we generated four independent samples containing tetramer[+] and tetramer[−] (Tconv) cells from two BDC2.5/IA[g7]-NP-treated mice for each sample. For TFH and TH0 cells, we prepared RNA from three independent TFH cell pools (CD4[+]/CD44[hi]/CXCR5[hi]/PD1[hi]) and TH0 cells (CD4[+]/CD44[−]/CXCR5[−]/PD1[−]), as a negative control. All samples were coming from three immunized mice each.

Total RNA was prepared from sorted cells using the RNeasy Plus Mini Kit (QIAGEN, Hilden, Germany) and used to prepare RNAseq libraries and sequencing. Libraries were prepared using the TruSeq Stranded mRNA Sample Prep Kit v2 according to the manufacturer's protocol (Illumina, San Diego, CA, USA). Briefly, 10–50 ng of total RNA was used for poly(A)-mRNA purification using streptavidin-coated magnetic beads, followed by fragmentation to ~300 bp. cDNA was synthesized using reverse transcriptase (SuperScript II, Invitrogen, Thermo Fisher Scientific, Waltham, MA, USA) and random primers. The second strand of the cDNA incorporated dUTP in place of dTTP. Double-stranded DNA (dsDNA) was further used for library preparation. dsDNA was subjected to A-tailing and ligation of the barcoded TruSeq adapters. All purification steps were performed using AMPure XP Beads (Beckman Coulter, Brea, CA, USA). Library amplification was performed by PCR using the primer cocktail supplied in the kit. Final libraries were analyzed using Agilent DNA 1000 chip to estimate the quantity and size distribution. They were then quantified by qPCR using the KAPA Library Quantification Kit (Kapa Biosystems, Roche, Basel, Switzerland) before amplification with Illumina's cBot. Libraries were loaded at a concentration of 2.75 pM onto the flowcell and sequenced 1×50 on Illumina's HiSeq 2500 to obtain 30–40 M reads.

## 10X scRNAseq

At least 5e4 fresh, alive cells were collected in DMEM media (Sigma-Aldrich) supplemented with 10% FBS (Hyclone) at 4°C and sent to CNAG-CRG for processing and sequencing. In short, cells were separated into nanoscale gel beads emulsions with a 10X barcode. Cell numbers and viability were assessed using a TC20 Automated Cell Counter (Bio-Rad Laboratories, Hercules, CA, USA), with a minimum target of 5000 cells. Later, cDNA sequencing libraries were produced using the NextGEM Single-cell 3' mRNA kit (v3.1; 10X Genomics) following the manufacturer's instructions. These steps involved GEM-RT clean-up, cDNA Amplification for 13 cycles, and cDNA quality control and quantification using the Agilent Bioanalyzer High Sensitivity chip (Agilent Technologies). Libraries were indexed by PCR using the PN-220103 Chromium i7 Sample Index Plate. Finally, sequencing was carried out on a NovaSeq 6000 sequencer (Illumina).

## ATACseq

For ATACseq (assay for transposase-accessible chromatin using sequencing), 5e4 cells were sorted in PBS and processed for library preparation as described by *Buenrostro et al., 2013*. Briefly, cells were lysed in cold lysis buffer (10 mM Tris-HCl, pH 7.4, 10 mM NaCl, 3 mM MgCl$_2$, and 0.1% IGEPAL CA-630), washed, and right after resuspended in transposase reaction mix (25 µL 2× TD buffer, 2.5 µL transposase [Illumina], and 22.5 µL nuclease-free water) and incubated for 30 min at 37°C. Next, library fragments were amplified using 1× NEB Next PCR master mix (New England BioLabs) and 1.25 µM of custom Nextera PCR primers forward and reverse. Libraries were rendered using the barcoded primers Ad1_noMX as forward and Ad2.1-6 as reverse and purified using a PCR cleanup kit (QIAGEN), yielding a final concentration of about 30 nM in 20 µL. Libraries were then analyzed on Bioanalyzer using an Agilent DNA High Sensitivity chip (Agilent Technologies, Santa Clara, CA, USA) to estimate the quantity and size distribution. Next, they were quantified by qPCR using the KAPA Library Quantification Kit before amplification with Illumina's cBot. Libraries were finally loaded at 3.33 pM onto the flowcell and sequenced 1×50 on Illumina's HiSeq 2500.

## 10X scMultiome (scRNAseq+scATACseq)

For 10X Multiome RNAseq+ATACseq, at least 5e5 fresh, alive cells were collected in DMEM media (Sigma-Aldrich) supplemented with 10% FBS (Hyclone) at 4°C, and then lysed and nuclei isolated.

Nuclei were transposated and adapter sequences added to DNA fragments. Nuclei were then processed for single-cell barcoding and library generation following the manufacturer's instructions (CG000338; 10X Genomics). Briefly, isolated nuclei were partitioned into Gel Bead-In-Emulsions to produce barcoded cDNA from poly-adenylated mRNA as described above, as well as barcoded DNA fragments, and processed for library amplification and sequencing on a NovaSeq 6000 sequencer (Illumina) as described above.

## ChIPseq

Chromatin immunoprecipitation (ChIP) and sequencing was performed for H3K4me3, H3K27me, and H3K27Ac bound DNA via ChIPseq. We used 1e6 cells. Cells were pooled from tetramer+ T-cells from BDC2.5/IA$^{g7}$-NP-treated mice (eight mice), and TFH (CD4$^+$/CD44$^{hi}$/CXCR5$^{hi}$PD1$^{hi}$), and TH0 cells (CD4$^+$/CD44$^-$/CXCR5$^-$/PD1$^-$) (extract from the same group of eight mice). In brief, cell dry pellets were fixed right after cell sorting with 10% PFA in DMEM (Sigma-Aldrich) supplemented with 10% FBS (Hyclone). Next, cells were lysed, sheared, and sonicated using an S220 Focused-ultrasonicator (Covaris, Woburn, MA, USA) (13 min, 105 W, 2% Duty Factor, 200 cycles). This was followed by over-night incubation with the precipitating antibody: 0.5 µL of H3K4me3 (Sigma), 0.5 µL of H3K27me3 (Cell Signaling, Danvers, MA, USA), and 2 µL of H3K27Ac (Abcam, Cambridge, United Carlsbad, CA, USA) and precipitated using Protein-A-Dynabeads (Abcam). RNA was cleared using RNAse A (QIAGEN) (1 hr at 65°C), and decrosslinking was performed overnight with proteinase K at 65°C. DNA was finally purified with Phenol-Chloroform and EtOH-precipitation. After validation by Bioanalyzer analysis quality control (Agilent Technologies), samples were sequenced. Libraries were prepared using the NEB Next Ultra DNA Library Prep kit (Illumina) following the manufacturer's protocol. Libraries were loaded at a concentration of 2.75 pM onto flowcells and were sequenced 1×50 on Illumina's HiSeq 2500.

## Methylome

For methylome analysis, genomic DNA was extracted using the DNeasy Blood and Tissue kit (QIAGEN) following the manufacturer's instructions. Samples were then sent to Beijing Genomics Institute (BGI, Shenzhen, China), once frozen, for sequencing and bioinformatics analysis. DNA was processed by whole-genome bisulfite sequencing. DNA was sonicated to a mean size of 250 bp using a Bioruptor (Diagenode, Belgium) and ends blunted by dA addition to the 3'-end. Finally, adapters were ligated to protect bisulfite conversions. Next, ligated DNA was bisulfite converted using an EZ DNA Methylation-Gold kit (ZYMO, Irvine, CA, USA). Unmethylated cytosines were converted into uracil, which after purification and amplification via PCR, were converted back to thymine. Finally, samples were sequenced at 2×150 bp using NovaSeq 6000 system (Illumina).

## Bioinformatic and statistical analyses

All fastq files obtained for each omics analysis were assessed for their quality control metrics before further downstream analysis using the FastQC tool. Sources for the indicated bioinformatic packages and tools are described further below.

### RNAseq

For bulk RNAseq analysis, fastq file reads were aligned using STAR to GRCm38.p6 mouse genome, and gene counts were obtained using Gencode M25 annotation release version simultaneously with the 'GeneCounts' STAR function. The resulting BAM files were processed into BigWig format for genomic tracks representation using SAMtools, deepTools, and trackViewer. Next, raw count values were processed and analyzed using R packages DESeq2 for normalization, scaling, and negative binomial distribution differential analysis. ggplot2 (tidyverse) was mainly used for graphics rendering purposes. Differential analysis log2 fold changes results were shrunk using 'apglm' to remove noise (*Zhu et al., 2019*).

### ATACseq and ChIPseq

For bulk ATACseq analysis, Illumina adapters and low-quality bases were first removed from fastq files reads using Trimmomatic. Next, reads were aligned to GRCm38.p6 mouse genome using Bowtie2, and duplicates were removed using Picard's 'MarkDuplicates'. Then peaks were called using MACS2

with a q-value cutoff of 0.05, read extension of 5'->3' of 150, and keeping duplicates as they had been removed previously ('-q 0.05 --nomodel --extsize 150 --keep-dup all'). The resulting BAM files were processed into BigWig format for genomic tracks representation using SAMtools, deepTools, and trackViewer.

For ATACseq analysis differential OCRs between samples were analyzed using DiffBind using BAM and peakset files. Briefly, we determined the overall background noise for each sample and then identified regions where the signal significantly exceeded noise, enabling accurate identification of peaks of enrichment controlling for false discovery rate (q-value<0.05). Differential OCRs between cell types were determined using likelihood ratio to compare read counts or signal intensity within OCRs across samples.

Differential peaks between ChIPseq samples were obtained using GSA (gene set analysis) from Partek. Peaks were annotated using 'annotatePeak' from the ChIPseeker package, using the UCSC mm10 reference included in org.Mm.eg.db and TxDb.Mmusculus.UCSC.mm10.knownGene R packages. Given that peak calling alone does not account for variations in the intensity of histone mark deposition, analysis of differential histone deposition includes both qualitative and quantitative assessments. Whereas qualitative assessment involves evaluating the overall pattern and distribution of the various histone marks, quantitative assessment measures the intensity and magnitude of histone mark deposition.

## Methylome

Upstream bioinformatic analysis of whole-genome methylome data was performed by the bioinformatics team at BGI. In short, sequencing data was first filtered to remove adaptor sequences and low-quality reads from raw reads. Filtered data was then mapped to the reference genome (mm10) by BSMAP, and duplication reads were removed. Regarding alignment quality metrics, the mapping rate and bisulfite conversion rate were measured for each sample. Only uniquely mapped data was used to get methylation data. Methylation level was determined by dividing the number of reads covering each mC by the total reads covering that cytosine. DMRs were identified by comparison between sample methylomes using windows that contained at least five CpG (CHG or CHH) sites with at least a twofold change in methylation level and Fisher's exact test p-value≤0.05. Adjacent DMRs would be considered interdependent and joined into one continuous DMR if all the regions were differentially methylated between samples. Otherwise, DMRs were identified as independent. Genomic tracks for methylome data were represented using the trackViewer R package.

## scRNAseq

10X scRNAseq data was demultiplexed, aligned, and counts measured using Cellranger software from 10X Genomics. In short, Cellranger 10X software first filter and trim low-quality reads, then align them to a reference genome using STAR. Next, UMI (reads) and cell barcodes are filtered, grouped, and counted. Cells are called and reported their gene expression in matrices based on RNA content for each cell barcode. We then performed the secondary analysis of gene expression using the Seurat R package, where we first discarded poor quality cells based on features counts and mitochondrial and ribosomal content. Then, data was normalized, scaled, and dimensionally reduced using PCA (principal component analysis) and UMAP. Finally, cells were clustered using K-means, and visualization and differential analysis were performed.

## scMultiome (scRNAseq+scATACseq)

10X single-cell multiomic data of simultaneous RNAseq and ATACseq was analyzed using 10X Genomics software Cellranger-arc. In this case, gene expression matrices from gene expression data are obtained like with Cellranger software (see the previous section). On the other hand, transposase accessibility data is adapter-removed and trimmed. Next, alignment is performed using the BWA-MEM algorithm, using a fixed insert size distribution, and duplicates removed. Afterward, peaks are called across all the cells to maximize the signal and then separated by barcode, obtaining peak-barcode matrices. Subsequently, gene expression and accessibility peaks matrices were combined and downstream-analyzed using Seurat and Signac packages. Like the scRNAseq pipeline, data were first filtered for poor-quality cells using features and peaks counts, mitochondrial content, nucleosome signal, and TSS enrichment. Later, RNA and ATAC data were normalized and scaled. Also,

each dataset was dimensionally reduced using PCA for RNA and LSI (Latent Semantic Indexing) for ATAC and UMAP. Multidimensional reduction of joint RNAseq and ATACseq data simultaneously was performed using the WNN algorithm from Seurat and clustered using K-means.

### Active enhancer prediction

ATACseq and H3K27ac-ChIPseq were used to predict potential active enhancer regions. Using the 'GenomicRanges' package, all peaks called for ATACseq overlapping peaks called for H3K27Ac deposition in the same sample, which were not in a promoter region (2 kb region upstream of TSS), were considered active enhancers.

### Chromosome views

We used the trackViewer R package to combine the information from RNAseq, ATACseq, ChIPseq, predicted active enhancers, and methylation data in linear plots representing gene tracks for specific genes. Alignment bam files for RNAseq, ATACseq, and ChIPseq were transformed to BigWig format using deepTools.

## Software and tools used for bioinformatic analyses

Bowtie2 (v2.4.2) (*Langmead and Salzberg, 2012*); BSMAP (v3.0) (*Xi and Li, 2009*); BWA (v0.0.7) (*Li and Durbin, 2010*); Cellranger (v6.0) (https://support.10xgenomics.com/single-cell-gene-expression/software/pipelines/latest/what-is-cell-ranger); Cellranger-arc (v2.0) (https://support.10xgenomics.com/single-cell-multiome-atac-gex/software/pipelines/latest/what-is-cell-ranger-arc); ChIPSeeker (v1.28.3) (*Yu et al., 2015*); clusterProfiler (v4.0.5) (*Wu et al., 2021*); deepTools (v3.5.0) (*Ramírez et al., 2014*); DESeq2 (v1.32.0) (*Love et al., 2014*); DiffBind (v3.2.7) (*Ross-Innes et al., 2012*); FastQC (http://www.bioinformatics.babraham.ac.uk/projects/fastqc/); FlowJo (v9) (https://www.flowjo.com); GenomicRanges (v1.44.0) (*Lawrence et al., 2013*); MACS2 (v2.2.7.1) (*Zhang et al., 2008*); org.Mm.eg.db (v3.13.0) (https://bioconductor.org/packages/release/data/annotation/html/org.Mm.eg.db.html); Partek Flow software (https://www.partek.com/partek-flow/); Picard (v2.25.0) (https://broadinstitute.github.io/picard/); R (v4.1.0), R Core Team (2020). — European Environment Agency, n.d. (https://www.eea.europa.eu/data-and-maps/indicators/oxygen-consuming-substances-in-rivers/r-development-core-team-2006); RStudio (v1.4.1103) (https://www.rstudio.com/); SAMtools (*Li et al., 2009*); Seurat (v4.0.3) (*Hao et al., 2021*); Signac (v1.3.0) (*Stuart et al., 2021*); STAR (v2.7.10a) (*Dobin et al., 2013*); tidyverse (v1.3.1) (https://www.tidyverse.org); trackViewer (v1.31.1) (*Ou and Zhu, 2019*); Trimmomatic (v.039) (*Bolger et al., 2014*).

## Statistical analyses

Statistical significance of the transcriptomic and epigenetic data was compared using the bioinformatic tools described above. Statistical significance for differences in the numbers of genes shared between different subsets was determined using the Chi-square test.

## Acknowledgements

We thank S Thiessen, J Erickson, J Fetsch, G Mendizabal, F Liu for technical contributions and/or animal care, Y Liu for flow cytometry, K Poon from the Nicole Perkins Microbial Communities Core for flow cytometry, P Nieto, E Mereu, G Lunazzi, and H Heyn from the Centre Nacional d'Anàlisi Genòmica (CNAG), I González and F Fernández from the Centre for Genomic Regulation (CRG), and S Wegener from the University of Calgary's Center for Health Genomics and Informatics, for library preparation and/or sequencing. This work was supported by Genome Canada and Genome Alberta (GAPP program), the Canadian Institutes of Health Research (CIHR) (FDN-353029, PJT-479040, PJT-479038, FRN-168480 [with JDRF], DT4-179512), the Praespero Foundation, the ISCIII and FEDER (PIE14/00027, PI15/0797), Ministerio de Ciencia e Innovación of Spain (MCIN; PID2021-125493OB-I00), Generalitat de Catalunya (SGR and CERCA Programmes) and Red Española de Supercomputación (RES, providing CSUC resources). J Garnica and P Solé were supported by a predoctoral studentship from FPU (MCIN). P Serra was an investigator of the Ramon y Cajal re-integration program and was supported by a JDRF Career Development Award.

# Additional information

### Competing interests
Pere Santamaria: P. Santamaria is founder, scientific officer and stockholder of Parvus Therapeutics and receives funding from the company. He also has a consulting agreement with Sanofi. The other authors declare that no competing interests exist.

### Funding

| Funder | Grant reference number | Author |
|---|---|---|
| Canadian Institutes of Health Research | FDN-353029 | Pere Santamaria |
| Ministry of Science and Innovation of Spain | PID2021-125493OB-I00 | Pau Serra<br>Pere Santamaria |
| Genome Alberta/Genome Canada | | Pere Santamaria |
| Canadian Institutes of Health Research | PJT-479040 | Pere Santamaria |
| Canadian Institutes of Health Research | PJT-479038 | Pere Santamaria |
| Canadian Institutes of Health Research | FRN-168480 (with JDRF) | Pere Santamaria |
| Canadian Institutes of Health Research | DT4-179512 | Pere Santamaria |
| Instituto de Salud Carlos III (ISCIII) | PIE14/00027 | Pere Santamaria |
| the Praespero Foundation | | Pere Santamaria |

The funders had no role in study design, data collection and interpretation, or the decision to submit the work for publication.

### Author contributions
Josep Garnica, Data curation, Formal analysis, Investigation, Visualization, Methodology, Writing – original draft; Patricia Sole, Investigation, Methodology; Jun Yamanouchi, Investigation; Joel Moro, Debajyoti Mondal, Cesar Fandos, Methodology; Pau Serra, Supervision; Pere Santamaria, Conceptualization, Resources, Supervision, Funding acquisition, Investigation, Methodology, Writing – original draft, Project administration, Writing – review and editing

### Author ORCIDs
Pere Santamaria ⬛ https://orcid.org/0000-0003-3469-1586

### Ethics
The experiments described herein were approved by the Cumming School of Medicine's Animal Care Committee at the University of Calgary Animal Care (protocol AC24-0053) and by the Animal Care Committee at Universitat de Barcelona (protocol 130/19).

Reviewer #1 (Public review): https://doi.org/10.7554/eLife.97665.3.sa1
Reviewer #2 (Public review): https://doi.org/10.7554/eLife.97665.3.sa2
Author response https://doi.org/10.7554/eLife.97665.3.sa3

# Additional files

### Supplementary files
• Supplementary file 1. Differential expression of T-regulatory type 1 (TR1)/T-follicular helper (TFH)/regulatory T-cell (Treg)-relevant genes in peptide-major histocompatibility complex class II (pMHCII)-nanoparticle (NP)-induced Tet+ vs. vaccine-induced TFH cells.

• MDAR checklist

## Data availability

The raw and processed data supporting the findings of this study are available at the Gene Expression Omnibus (GEO) database. Accession numbers are as follows: GSE173681 (bulk RNA-seq data); GSE182636 (scRNA-seq data); and GSE248152 (ATAC-seq, ChIP-seq, methylome, and single-cell multiome data).

The following datasets were generated:

| Author(s) | Year | Dataset title | Dataset URL | Database and Identifier |
|---|---|---|---|---|
| Garnica J, Sole P, Yamanouchi J, Moro J, Mondal D, Fandos C, Serra P, Santamaria P | 2023 | Transcriptional profiline of murine and human T cell populations | https://www.ncbi.nlm.nih.gov/geo/query/acc.cgi?acc=GSE173681 | NCBI Gene Expression Omnibus, GSE173681 |
| Garnica J, Sole P, Yamanouchi J, Moro J, Mondal D, Fandos C, Serra P, Santamaria P | 2023 | Transcriptional single-cell study of Nanoparticle-derived TR1-like cells in NOD.Prdm1 flox/floxCD4-Cre+ mice. | https://www.ncbi.nlm.nih.gov/geo/query/acc.cgi?acc=GSE182636 | NCBI Gene Expression Omnibus, GSE182636 |
| Garnica J, Sole P, Yamanouchi J, Moro J, Mondal D, Fandos C, Serra P, Santamaria P | 2024 | ATAC-seq, ChIP-seq, methylome, and single-cell multiome data | https://www.ncbi.nlm.nih.gov/search/all/?term=GSE248152 | NCBI Gene Expression Omnibus, GSE248152 |

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
