## [Editor Report · eLife Assessment]

This study provides **important** information on pre-existing epigenetic modification in T cell plasticity. The evidence supporting the conclusions is **compelling**, supported by comprehensive transcriptional and epigenetic analyses. The work will be of interest to immunologists and colleagues studying transcriptional regulation.

---

## [Referee Report · Reviewer #1 (Public review)]

Summary:

Dr. Santamaria's group previously utilized antigen-specific nanomedicines to induce immune tolerance in treating autoimmune diseases. The success of this therapeutic strategy has been linked to expanded regulatory mechanisms, particularly the role of T-regulatory type-1 (TR1) cells. However, the differentiation program of TR1 cells remained largely unclear. Previous work from the authors suggested that TR1 cells originate from T follicular helper (TFH) cells. In the current study, the authors aimed to investigate the epigenetic mechanisms underlying the transdifferentiation of TFH cells into IL-10-producing TR1 cells. Specifically, they sought to determine whether this process involves extensive chromatin remodeling or is driven by pre-existing epigenetic modifications. Their goal was to understand the transcriptional and epigenetic changes facilitating this transition and to explore the potential therapeutic implications of manipulating this pathway.

The authors successfully demonstrated that the TFH-to-TR1 transdifferentiation process is driven by pre-existing epigenetic modifications rather than extensive new chromatin remodeling. The comprehensive transcriptional and epigenetic analyses provide robust evidence supporting their conclusions.

Strengths:

(1) The study employs a broad range of bulk and single-cell transcriptional and epigenetic tools, including RNA-seq, ATAC-seq, ChIP-seq, and DNA methylation analysis. This comprehensive approach provides a detailed examination of the epigenetic landscape during the TFH-to-TR1 transition.

(2) The use of high-throughput sequencing technologies and sophisticated bioinformatics analyses strengthens the foundation for the conclusions drawn.

(3) The data generated can serve as a valuable resource for the scientific community, offering insights into the epigenetic regulation of T cell plasticity.

(4) The findings have significant implications for developing new therapeutic strategies for autoimmune diseases, making the research highly relevant and impactful.

Weaknesses:

(1) While the study focuses on transcriptional and epigenetic analyses, the authors are currently undertaking efforts to validate these findings functionally. Ongoing research aims to further explore the roles of key transcription factors in the TFH-to-TR1 transition, reflecting the authors' commitment to building on the insights gained from this study.

(2) The identification of key transcription factors and epigenetic marks is a strong foundation for future work. The authors are actively investigating how these factors drive chromatin remodeling, which will enhance the mechanistic understanding of the TFH-to-TR1 process in future studies.

(3) Although the study provides a valuable snapshot of the epigenetic landscape, the authors are pursuing additional research to assess the dynamics of these changes over time. These ongoing efforts will contribute to a deeper understanding of the stability and progression of the observed epigenetic modifications.

Comments on revised version:

The authors have effectively discussed and addressed all previously raised questions. There are no further concerns.

---

## [Referee Report · Reviewer #2 (Public review)]

Summary:

This study, based on their previous findings that TFH cells can be converted into TR1 cells, conducted a highly detailed and comprehensive epigenetic investigation to answer whether TR1 differentiation from TFH is driven by epigenetic changes. Their evidence indicated that the downregulation of TFH-related genes during the TFH to TR1 transition depends on chromatin closure, while the upregulation of TR1-related genes does not depend on epigenetic changes.

Strengths:

A significant advantage of their approach lies in its detailed and comprehensive assessment of epigenetics. Their analysis of epigenetics covers chromatin open regions, histone modifications, DNA methylation, and using both single-cell and bulk techniques to validate their findings. As for their results, observations from different epigenetic perspectives mutually supported each other, lending greater credibility to their conclusions. This study effectively demonstrates that 1. the TFH-to-TR1 differentiation process is associated with massive closure of OCRs, and 2. the TR1-poised epigenome of TFH cells is a key enabler of this transdifferentiation process. Considering the extensive changes in epigenetic patterns involved in other CD4+ T lineage commitment processes, the similarity between TFH and TR1 in their epigenetics is intriguing.

They performed correlation analysis to answer the association between "pMHC-NP-induced epigenetic change" and "gene expression change in TR1". Also, they have made their raw data publicly available, providing a comprehensive epigenomic database of pMHC-NP induced TR1 cells. This will serve as a valuable reference for future research.

Weaknesses:

A major limitation is that this study heavily relies on a premise from the previous studies performed by the same group on pMHC-NP-induced T cell responses. This significantly limits the relevance of their conclusion to a broader perspective. Specifically, differential OCRs between Tet+ and naïve T cells were limited to only 821, as compared to 10,919 differential OCRs between KLH-TFH and naïve T cells (Fig. 2A), indicating that the precursors and T cell clonotypes that responded to pMHC-NP were extremely limited. I acknowledge that this limitation has been added and discussed in the Discussion section of the revised manuscript.

---

## [Author Response]

The following is the authors’ response to the original reviews.

**Reviewer #1 (Public Review):**
Summary:Dr. Santamaria's group previously utilized antigen-specific nanomedicines to induce immune tolerance in treating autoimmune diseases. The success of this therapeutic strategy has been linked to expanded regulatory mechanisms, particularly the role of T-regulatory type-1 (TR1) cells. However, the differentiation program of TR1 cells remained largely unclear. Previous work from the authors suggested that TR1 cells originate from T follicular helper (TFH) cells. In the current study, the authors aimed to investigate the epigenetic mechanisms underlying the transdifferentiation of TFH cells into IL-10-producing TR1 cells. Specifically, they sought to determine whether this process involves extensive chromatin remodeling or is driven by preexisting epigenetic modifications. Their goal was to understand the transcriptional and epigenetic changes facilitating this transition and to explore the potential therapeutic implications of manipulating this pathway.The authors successfully demonstrated that the TFH-to-TR1 transdifferentiation process is driven by pre-existing epigenetic modifications rather than extensive new chromatin remodeling. The comprehensive transcriptional and epigenetic analyses provide robust evidence supporting their conclusions.Strengths:(1) The study employs a broad range of bulk and single-cell transcriptional and epigenetic tools, including RNA-seq, ATAC-seq, ChIP-seq, and DNA methylation analysis. This comprehensive approach provides a detailed examination of the epigenetic landscape during the TFH-to-TR1 transition.(2) The use of high-throughput sequencing technologies and sophisticated bioinformatics analyses strengthens the foundation for the conclusions drawn.(3) The data generated can serve as a valuable resource for the scientific community, offering insights into the epigenetic regulation of T-cell plasticity.(4) The findings have significant implications for developing new therapeutic strategies for autoimmune diseases, making the research highly relevant and impactful.We thank the reviewer for providing constructive feedback on the manuscript.Weaknesses:(1) While the scope of this study lies in transcriptional and epigenetic analyses, the conclusions need to be validated by future functional analyses.

We fully agree with the reviewer’s suggestion. We have added the following text to the Discussion to address this concern: “The current study provides a foundational understanding of how the epigenetic landscape of TFH cells evolves as they transdifferentiate into TR1 progeny in response to chronic ligation of cognate TCRs using pMHCII-NPs. Our current studies focus on functional validation of these observations, by carrying out extensive perturbation studies of the TFH-TR1 transdifferentiation pathway in conditional transcription factor gene knock-out mice. In these ongoing studies, genes coding for a series of transcription factors expressed along the TFH-TR1 pathway are selectively knocked out in T cells, to ascertain (i) the specific roles of key transcription factors in the various cell conversion events and transcriptional changes that take place along the TFH-TR1 cell axis; (ii) the roles that such transcription factors play in the chromatin re-modeling events that underpin the TFH-TR1 transdifferentiation process; and (iii) the effects of transcription factor gene deletion on phenotypic and functional readouts of TFH and regulatory T cell function.”

(2) This study successfully identified key transcription factors and epigenetic marks. How these factors mechanistically drive chromatin closure and gene expression changes during the TFH-to-TR1 transition requires further investigation.

Agreed. Please see our response to point #1 above.

(3) The study provides a snapshot of the epigenetic landscape. Future dynamic analysis may offer more insights into the progression and stability of the observed changes.

We have previously shown that the first event in the pMHCII-NP-induced TFH-TR1 transdifferentiation process involves proliferation of cognate TFH cells in the splenic germinal centers. This event is followed by immediate transdifferentiation of the proliferated TFH cells into transitional and terminally differentiated TR1 subsets. Although the snapshot provided by our single cell studies reported herein documents the simultaneous presence of the different subsets composing the transdifferentiation pathway at any given time point, the transdifferentiation process itself is extremely fast, such that proliferated TFH cells already transdifferentiate into TR1 cells after a single pMHCII-NP dose (Sole et al., 2023a). This makes it extremely challenging to pursue dynamic experiments. Notwithstanding this caveat, ongoing studies of cognate T cells post treatment withdrawal, coupled to single cell studies of the TFHTR1 pathway in transcription factor gene knockout mice exhibiting perturbed transdifferentiation processes are likely to shed light into the progression and stability of the epigenetic changes reported herein.

To address this limitation in the manuscript, we have added the following paragraph to the Discussion: “Although the snapshot provided by our single cell studies reported herein documents the simultaneous presence of the different subsets composing the TFH-TR1 cell pathway upon the termination of treatment, the transdifferentiation process itself is extremely fast, such that proliferated TFH cells already transdifferentiate into TR1 cells after a single pMHCII-NP dose (6). This makes it extremely challenging to pursue dynamic experiments. Notwithstanding this caveat, ongoing studies of cognate T cells post treatment withdrawal, coupled to single cell studies of the TFH-TR1 pathway in transcription factor gene knockout mice exhibiting perturbed transdifferentiation processes are likely to shed light into the progression and stability of the epigenetic changes reported herein”.

**Reviewer #1 (Recommendations for the authors):**
The authors may consider the following suggestions to improve this study:(1) The authors may include a brief background on type 1 diabetes and the model involving BDC2.5 T cells to provide context for readers who may not be familiar with these aspects.

We have added this information to the first paragraph in the Results section: “BDC2.5mi/I-Ag7-specific CD4+ T cells comprise a population of autoreactive T cells that contribute to the progression of spontaneous autoimmune diabetes in NOD mice. The size of this type 1 diabetes-relevant T cell specificity is small and barely detectable in untreated NOD mice, but treatment with cognate pMHCII-NPs leads to the expansion and formation of antidiabetogenic TR1 cells that retain the antigenic specificity of their precursors (3). As a result, treatment of hyperglycemic NOD mice with these compounds results in the reversal of type 1 diabetes (3).”

(2) It is understandable that further biological and functional experiments are beyond the scope of this paper, but it would be of interest to know how the authors envision future studies based on the transcriptional and epigenetic information obtained thus far.

We have added the following text to the Discussion section: “The current study provides a foundational understanding of how the epigenetic landscape of TFH cells evolves as they transdifferentiate into TR1 progeny in response to chronic ligation of cognate TCRs using pMHCII-NPs. Our current studies focus on functional validation of these observations, by carrying out extensive perturbation studies of the TFH-TR1 transdifferentiation pathway in conditional transcription factor gene knock-out mice. In these ongoing studies, genes coding for a series of transcription factors expressed along the TFH-TR1 pathway are selectively knocked out in T cells, to ascertain (i) the specific roles of key transcription factors in the various cell conversion events and transcriptional changes that take place along the TFH-TR1 cell axis; (ii) the roles that such transcription factors play in the chromatin re-modeling events that underpin the TFH-TR1 transdifferentiation process; and (iii) the effects of transcription factor gene deletion on phenotypic and functional readouts of TFH and regulatory T cell function.”

(3) The authors may consider adjusting figures where genes are crowded or difficult to read due to small font size.

Figures with crowded text have been modified to facilitate reading.

**Reviewer #2 (Public Review):**
Summary:This study, based on their previous findings that TFH cells can be converted into TR1 cells, conducted a highly detailed and comprehensive epigenetic investigation to answer whether TR1 differentiation from TFH is driven by epigenetic changes. Their evidence indicated that the downregulation of TFH-related genes during the TFH to TR1 transition depends on chromatin closure, while the upregulation of TR1-related genes does not depend on epigenetic changes.Strengths:(1) A significant advantage of their approach lies in its detailed and comprehensive assessment of epigenetics. Their analysis of epigenetics covers chromatin open regions, histone modifications, DNA methylation, and using both single-cell and bulk techniques to validate their findings. As for their results, observations from different epigenetic perspectives mutually supported each other, lending greater credibility to their conclusions. This study effectively demonstrates that (1) the TFH-to-TR1 differentiation process is associated with massive closure of OCRs, and (2) the TR1-poised epigenome of TFH cells is a key enabler of this transdifferentiation process. Considering the extensive changes in epigenetic patterns involved in other CD4+ T lineage commitment processes, the similarity between TFH and TR1 in their epigenetics is intriguing.(2) They performed correlation analysis to answer the association between "pMHC-NPinduced epigenetic change" and "gene expression change in TR1". Also, they have made their raw data publicly available, providing a comprehensive epigenomic database of pMHC-NPinduced TR1 cells. This will serve as a valuable reference for future research.

We thank the reviewer for his/her constructive feedback and suggestions for improvement of the manuscript.

Weaknesses:(1) A major limitation is that this study heavily relies on a premise from the previous studies performed by the same group on pMHC-NP-induced T-cell responses. This significantly limits the relevance of their conclusion to a broader perspective. Specifically, differential OCRs between Tet+ and naïve T cells were limited to only 821, as compared to 10,919 differential OCRs between KLH-TFH and naïve T cells (Figure 2A), indicating that the precursors and T cell clonotypes that responded to pMHC-NP were extremely limited. This limitation should be clearly discussed in the Discussion section.

We agree that this study focuses on a very specific, previously unrecognized pathway discovered in mice treated with pMHCII-NPs. Despite this apparent narrow perspective, we now have evidence that this is a naturally occurring pathway that also develops in other contexts (i.e., in mice that have not been treated with pMHCII-NPs). Furthermore, this pathway affords a unique opportunity to further understand the transcriptional and epigenetic mechanisms underpinning T cell plasticity; the findings reported can help guide/inform not only upcoming translational studies of pMHCII-NP therapy in humans, but also other research in this area.

We have added the following text to the Discussion to address this limitation: “Although the TFH-TR1 transdifferentiation was discovered in mice treated with pMHCII-NPs, we now have evidence that this is a naturally occurring pathway that also develops in other contexts (i.e., in mice that have not been treated with pMHCII-NPs). Importantly, the discovery of this transdifferentiation process affords a unique opportunity to further understand the transcriptional and epigenetic mechanisms underpinning T cell plasticity; the findings reported here can help guide/inform not only upcoming translational studies of pMHCII-NP therapy in humans, but also other research in this area”.

We acknowledge that, in the bulk ATAC-seq studies, the differences in the number of OCRs found in tetramer+ cells or KLH-induced TFH cells vs. naïve T cells may be influenced by the intrinsic oligoclonality of the tetramer+ T cell pool arising in response to repeated pMHCII-NP challenge (Sole et al., 2023a). However, we note that our scATAC-seq studies of the tetramer+ T cell pool found similar differences between the oligoclonal tetramer+ TFH subpool and its (also oligoclonal) tetramer+ TR1 counterparts (i.e., substantially higher number of OCRs in the former vs. the latter relative to naïve T cells).

This has been clarified in the revised version of the manuscript, by adding the following text to the last paragraph of the Results subsection entitled “Contraction of the chromatin in pMHCII-NP-induced Tet+ vs. TFH cells at the bulk level”: “We acknowledge that, in the bulk ATAC-seq studies, the differences in the number of OCRs found in tetramer+ cells or KLHinduced TFH cells vs. naïve T cells may be influenced by the intrinsic oligoclonality of the tetramer+ T cell pool arising in response to repeated pMHCII-NP challenge (6). However, we note that scATAC-seq studies of the tetramer+ T cell pool found similar differences between the oligoclonal tetramer+ TFH subpool and its (also oligoclonal) tetramer+ TR1 counterparts (i.e., substantially higher number of OCRs in the former vs. the latter relative to naïve T cells)”.

(2) This article uses peak calling to determine whether a region has histone modifications, claiming that the regions with histone modifications in TFH and TR1 are highly similar. However, they did not discuss the differences in histone modification intensities measured by ChIP-seq. For example, as shown in Figure 6C, IL10 H3K27ac modification in Tet+ cells showed significantly higher intensity than KLH-TFH, while in this article, it may be categorized as "possessing same histone modification region". This will strengthen their conclusions.

We appreciate your suggestion to discuss differences in histone modification intensities as measured by ChIP-seq. However, we respectfully disagree with the reviewer’s interpretation of these data.

Our study primarily focuses on the identification of epigenetic similarities and differences between pMHCII-NP-induced tetramer+ cells and KLH-induced TFH cells relative to naive T cells. The outcome of direct comparisons of histone deposition (ChIP-seq) between these cell types is summarized in the lower part of Figure 4B and detailed in Datasheet 5. Throughout this section, we mention the number of differentially enriched regions, their overlap with OCRs shared between tetramer+ TFH and tetramer+ TR1 cells based on scATAC-seq data, and the associated genes. Clearly, the epigenetic modifications that TR1 cells inherit from TFH cells were acquired by TFH cells upon differentiation from naïve T cell precursors.

Regarding the specific point raised by the reviewer on differences in the intensity of the H3K27Ac peaks linked to *Il10* in Figure 6C, we note that the genomic tracks shown are illustrative. Thorough statistical analyses involving signal background for each condition and p-value adjustment did not support differential enrichment for H3K27Ac deposition around the *Il10* gene between pMHCII-NP-induced tetramer+ T cells and KLH-induced TFH cells.

This has now been clarified by adding the following text to the end of the Results subsection entitled ”H3K4me3, H3K27me3 and H3K27ac marks in genes upregulated during the TFH-to-TR1 cell conversion are already in place at the TFH cell stage”: “We note that, although in the representative chromosome track views shown in Fig. 6C there appear to be differences in the intensity of the peaks, thorough statistical analyses involving signal background for each condition and p-value adjustment did not support differential enrichment for histone deposition around the Il10 gene between pMHCII-NP-induced tetramer+ T cells and KLH-induced TFH cells.”

We have also clarified this in the corresponding section of the Methods section (“ATACseq and ChIP-seq” under “Bioinformatic and Statistical Analyses”): “Given that peak calling alone does not account for variations in the intensity of histone mark deposition, analysis of differential histone deposition includes both qualitative and quantitative assessments. Whereas qualitative assessment involves evaluating the overall pattern and distribution of the various histone marks, quantitative assessment measures the intensity and magnitude of histone mark deposition.”

(3) Last, the key findings of this study are clear and convincing, but some results and figures are unnecessary and redundant. Some results are largely a mere confirmation of the relationship between histone marks and chromatin status. I propose to reduce the number of figures and text that are largely confirmatory. Overall, I feel this paper is too long for its current contents.

We understand your concern about the potential redundancy of some results and figures. Our aim in including these analyses was to provide a comprehensive understanding of the intricate relationships between epigenetic features and transcriptomic differences. We believe that a detailed examination of these relationships is crucial for several reasons: (i) the breadth of the data allows for a thorough exploration of the relationships between histone marks, open chromatin status and transcriptional differences. This comprehensive approach helps to ensure that our conclusions are robust and well-supported; (ii) some of the results that may appear confirmatory are, in fact, important for validating and reinforcing the consistency of our findings across different contexts. These details are intended to provide a nuanced understanding of the interactions between epigenetic features and gene expression; and (iii) By presenting a detailed analysis, we aim to offer a solid foundation for future research in this area. The extensive data presented will serve as a valuable resource for others in the field who may seek to build on our findings.

That said, we have carefully reviewed the manuscript to identify and streamline elements that might be perceived as overly redundant, while retaining the depth of analysis that we believe is essential.

**Reviewer #2 (Recommendations for the authors):**
(1) In Figure 1E, the text states "94% (n=217/231) of the genes associated with chromatin regions that had closed during the TFH-TR1 conversion,", but n=231 do not match with n=1820 provided in Figure 1D as downregulated genes. This is one of the examples that do not match numbers among figures or lack sufficient explanations. Please check those numbers carefully and add some sentences if necessary.

We note that the text referring to Figure 1D describes the total number of differentially expressed genes between Tet+ TR1 and Tet+ TFH cells using the scMultiome dataset (n = 2,086 genes downregulated in the former vs. the latter; and n = 266 genes upregulated in the former vs. the latter). The text in the paragraph that follows (referring to Figure 1E) focuses exclusively on the genes that had closed chromatin regions during the TFH-to-TR1 conversion, to ascertain whether or not chromatin closure was indeed associated with such gene downregulation.

We have modified the first sentence in the paragraph referring to Figure 1E to clarify this point for the reader: “Further analyses focusing on the genes that had closed chromatin regions during the TFH-to-TR1 conversion, confirmed…”